# Inter-annual variations of wet deposition in Beijing during 2014-2017: implications of below-cloud scavenging of inorganic aerosols

Baozhu Ge[1,5*], Danhui Xu[1], Oliver Wild[2], Xuefeng Yao[3], Junhua Wang[1,4], Xuechun Chen[1], Qixin Tan[1,4], Xiaole Pan[1], Zifa Wang[1,4,5*]

[1] State Key Laboratory of Atmospheric Boundary Layer Physics and Atmospheric Chemistry (LAPC), Institute of Atmospheric Physics (IAP), Chinese Academy of Sciences (CAS), Beijing 100029, China

[2] Lancaster Environment Centre, Lancaster University, LA1 4YQ, United Kingdom

[3] PLA 96941 Army, Beijing 102206, China

[4] University of Chinese Academy of Sciences, Beijing, 100049, China

[5] Center for Excellence in Regional Atmospheric Environment, Institute of Urban Environment, Chinese Academy of Sciences, Xiamen 361021, China

*Correspondence to:* Baozhu Ge (gebz@mail.iap.ac.cn) and Zifa Wang (zifawang@mail.iap.ac.cn)

**Abstract**

Wet scavenging is an efficient pathway for the removal of particulate matter (PM) from the atmosphere. High levels of PM have been a major cause of air pollution in Beijing but have decreased sharply under the Air Pollution Prevention and Control Action Plan launched in 2013. In this study, four years of observations of wet deposition have been conducted using a sequential sampling technique to investigate the detailed variation in chemical components through each rainfall event. We find that the major ions, $SO_4^{2-}$, $Ca^{2+}$, $NO_3^-$ and $NH_4^+$, show significant decreases over the 2013-2017 period (decreasing by 39%, 35%, 12% and 25%, respectively), revealing the impacts of the Action Plan. An improved method of estimating the below-cloud scavenging proportion based on sequential sampling is developed and implemented to estimate the contribution of below-cloud and in-cloud wet deposition over the four-year period. Overall, the below-cloud scavenging plays a dominant role to the wet deposition of four major ions at the beginning of the Action Plan. The contribution of below-cloud scavenging for $Ca^{2+}$, $SO_4^{2-}$ and $NH_4^+$ decreases from above 50% in 2014 to below 40% in 2017. This suggests that the Action Plan has mitigated PM pollution in the surface layer and hence decreased scavenging due to the washout process. In contrast, we find little change in the annual volume weighted average concentration for $NO_3^-$ where the contribution from below-cloud scavenging remains at ~44% over the period 2015-2017. While highlighting the importance of different wet scavenging processes, this paper presents a unique new perspective on the effects of the Action Plan and clearly identifies oxidized nitrogen species as a major target for future air pollution controls.

**Key words:** wet scavenging, below-cloud, in-cloud, deposition, $PM_{2.5}$

## 1 Introduction

Atmospheric wet deposition is a key removal pathway for air pollutants and is governed by two main processes: in-cloud and below-cloud scavenging (Goncalves et al., 2002; Andronache, 2003, 2004a; Henzing et al., 2006; Sportisse, 2007; Feng, 2009; Wang et al., 2010; Zhang et al., 2013). The below-cloud scavenging process depends both on the characteristics of the rain (snow), including the raindrop size distribution and rainfall rate, and on the chemical nature of the particles and their concentration in the atmosphere (Chate et al., 2003). Previously, below-cloud scavenging was thought to be less important than in-cloud processes and was simplified or even ignored in many global and regional chemical transport models (CTMs) (Barth et al., 2000;Tang et al., 2005;ENVIRON.Inc, 2005;Textor et al., 2006;Bae et al., 2010). However, more recent extensive research on wet scavenging has found that precipitation, even light rain, can remove 50-80% of the number or mass concentration of below-cloud aerosols, and this is supported by both field measurements and semi-empirical parameterizations of below-cloud scavenging in models (Andronache, 2004b;Zhang et al., 2004;Wang et al., 2014). Xu et al. (2017;2019) studied the below-cloud scavenging mechanism based on the simultaneous measurement of aerosol components in rainfall and in the air in Beijing. They found that below-cloud scavenging coefficients for $PM_{2.5}$ widely used in CTMs ($\sim 10^{-5}$-$10^{-6}$) were 1-2 orders of magnitude lower than estimates from observations (at the range of $10^{-4}$-$10^{-5}$ for $SO_4^{2-}$, $NO_3^-$ and $NH_4^+$, respectively). This implies that the simulated below-cloud scavenging of aerosols might be significantly underestimated. This could be one reason for the underestimation of $SO_4^{2-}$ and $NO_3^-$ wet deposition in regional models of Asia reported in phase II and III of the Model Inter-Comparison Study for Asia (MICS-Asia) (Wang et al., 2008; Itahashi et al., 2020; Ge et al., 2020) and in global model assessments by the Task Force on Hemispheric Transport of Atmospheric Pollutants (TF-HTAP) (Vet et al., 2014), in addition to the other sources of model uncertainties (Chen et al., 2019; Tan et al., 2020; Kong et al., 2020), such as emissions, chemical transformation and changes in other ambient compounds of sulfur and nitrogen. Bae et al. (2012) added a new below-cloud scavenging parameterization scheme in the CMAQ model and improved the simulation

of aerosol wet deposition fluxes in East Asia by as much as a factor of two compared with observations. The below-cloud scavenging process is critical not only for wet deposition but also for the concentration of aerosols in the air and it should be represented appropriately in CTM simulations.

It is important to recognize the contribution of below-cloud scavenging to total wet deposition. However, many studies have found that it is difficult to separate the two wet scavenging processes based on measurement methods alone (Huang et al., 1995;Wang and Wang, 1996;Goncalves et al., 2002;Bertrand et al., 2008;Xu et al., 2017). A commonly used approach to separating below-cloud scavenging from total wet deposition is through sequential sampling (Aikawa et al., 2014;Ge et al., 2016;Aikawa and Hiraki, 2009;Wang et al., 2009;Quyang et al., 2015; Xu et al., 2017). In this way, precipitation composition during different stages of a rainfall event can be investigated separately in the lab after sampling. The chemical components in later increments of rainfall are thought to be less influenced by the below-cloud scavenging process than by the in-cloud scavenging process (Aikawa et al., 2014;2009). Xu et al. (2017) applied this approach to summer rainfall in Beijing in 2014 and found that more than 50% of deposited sulfate, nitrate and ammonium ions were from below-cloud scavenging. In this study, an innovated method based on exponential curve to chemical ions in rainfall by sequential sampling is developed and implemented to estimate the ratio of below-cloud to in-cloud wet deposition in Beijing over the four-year period between 2014 and 2017. Together with $PM_{2.5}$ concentration measurements, the below-cloud scavenging effects of the decreasing air pollutants at near-surface due to the Air Pollution Prevention and Control Action Plan (Action Plan) launched in 2013 (State Council of the People's Republic of China, 2019) is also investigated to explore the implications of the Action Plain to the precipitation chemistry.

**2 Data and methods**

**2.1 Measurement site and sampling methodology**

The measurement site is located on the roof of a two-floor building at the Institute of Atmospheric Physics tower site (IAP-tower, 39° 58′ 28″ N, 116° 22′ 1″ E) in northern Beijing. It is a typical urban site between the 3$^{rd}$ and 4$^{th}$ ring roads and lying close to

the Badaling expressway (Xu et al., 2017;2019;Sun et al., 2015). Four years of Inter-annual observations of each rainfall event were conducted at this site. Sequential sampling of each rainfall event is employed to catch the evolution of precipitation composition during each event. To investigate the detailed variation in the concentration of different chemical components in precipitation, especially the sharp changes occurring during the onset of rainfall, high resolution sampling of rainfall at 1 mm sequential increments was performed using an automatic wet-dry sampler. The rainwater collector uses a circular polyethylene board with a 30 cm diameter and collects up to eight fractions. About 70 ml of rainwater is collected for each of the first seven fractions and the rest of the rainfall is collected in the eighth fraction. For example, if there is 12 mm rainfall volume in a precipitation event, 1 mm sequential rainfall is collected in each of the first 7 fractions with the rest of 5 mm in the eighth fraction. Rainfall events where eight fractions are collected and identified as full events, and those with fewer than eight fractions are characterized as incomplete events. Manual sampling methods were used to collect more than eight fractions during heavy rainfall, and these are characterized as extended events. During 2014-2017, a total of 104 precipitation events, which is almost 690 precipitation samples, were collected. Of the total number of precipitation events, 33 events (32%) were discarded from the sequential sampling analysis due to low rainfall amounts (<8 mm), which cannot satisfy the rules of full events. Altogether, 69 full events including 6 extended events were recorded over the 2014-2017 period in Beijing, as 15, 16, 20 and 18 events at each year, respectively. The rainfall volume of the eighth fraction of these 69 full events varied from 1 mm to 55.9 mm.

After collection, all samples are refrigerated at 0-4 ºC and analyzed at the Key Laboratory for Atmospheric Chemistry, Chinese Academy of Meteorological Sciences (CAMS) within one month, following the procedure used for the Acid Rain Monitoring Network run by the China Meteorological Administration (CMA-ARMN) (Tang et al., 2007;2010). Nine ions that include four anions ($SO_2^{4-}$, $NO_3^-$, $Cl^-$ and $F^-$) and five cations ($NH_4^+$, $Na^+$, $K^+$, $Ca^{2+}$ and $Mg^{2+}$) are detected using ion chromatography (IC, Dionex 600, USA). Their relative standard deviations in reproducibility tests are less than 5%.

Quality assurance is carried out using routine standard procedure of blind sample inter-
comparison organized by CMA (Tang et al., 2010). Quality control is conducted by
assessment of the anion-cation balance and by comparison of the calculated and
measured conductivity. A more detailed description of the procedure can be found in
Ge at al. (2016) and Xu et al. (2017).

**2.2 Aerosol measurements**

Aerosol mass concentration is recorded in routine measurements for the observation
network of the China National Environmental Monitoring Center (CNEMC). $PM_{2.5}$
concentrations are used from the Olympic Park station, a monitoring station located 3
km to the northeast of the IAP-tower sampling site. In addition, an Ambient Ion
Monitor-Ion Chromatograph (AIM-IC) developed by URG Corp., Chapel Hill, NC and
Dionex Inc., Sunnyvale, CA, is used to measure $PM_{2.5}$ composition at the sampling site
between 2014 and 2017. This instrument includes a sample collection unit (URG 9000-
D) for collection of water-soluble gases and particles in aqueous solution and a sample
analysis unit (two ion chromatographs, Dionex ICS-2000 and ICS-5000) for analysis
of both anions and cations. The limit of detection of AIM-IC is 0.08 mg/m$^3$ for $NH^{4+}$
and 0.1 mg/m$^3$ for the other ions. Aerosol mass concentrations and composition are both
measured at 1 h time resolution. Detailed descriptions of the AIM-IC instrumentation
can be found in Malaguti et al. (2015) and Markovic et al. (2012). The average
concentration of aerosols in the 6 h before each rainfall event is calculated to reflect the
air pollution conditions before the event. For comparisons, the yearly average
concentration of aerosols has been calculated to represent the normal conditions.

**2.3 Estimation of below cloud scavenging**

Previous studies have shown that the concentration of chemical ions in precipitation
decreases through the progression of a rainfall event and eventually stabilizes at low
levels (Aikawa and Hiraki, 2009;2014;Ge et al., 2016;Xu et al., 2017). The in-cloud
and below-cloud scavenging contributions to total wet deposition are estimated based
on the assumption that the concentrations in later increments can be attributed to
scavenging by rainout only. According to (Seinfeld and Pandis, 2006), species can be
incorporated into cloud and raindrops inside the raining cloud and this process
determine the initial concentration of raindrops before they start falling below the cloud
base. In this stage, despite of the efficient process of the nucleation scavenging in cloud,
the total mass of aerosol in cloud is almost stable due to the slow process of interstitial
aerosol collection by cloud droplets which is the determination process to aerosol mass.
That is to say, the initial concentration of raindrops in cloud is well mixed and can be
considered as a stable statue during the whole rainfall event. That is why many
observations in different regions (Aikawa *et al*., 2009; 2014; Wang et al., 2009; Quyang
et al., 2015; Xu et al., 2017) reported that the chemical components in a rainfall event
show a decayed variation with the increase of precipitation amount and eventually tends
to a stable and low concentration level. The assumption in this study as well as the
previous studies is based on this fact. It does not mean the below-cloud and in-cloud
scavenging occur in sequence. But, instead, the two processes have been mixed in all
stage of the rainfall event with the below-cloud scavenging contributed more in
beginning fraction and the in-cloud scavenging contributed more in the later fraction
due to the depletion of the air pollutants below cloud by washout.
This assumption relies on the efficient scavenging of air pollutants below cloud
through the evolution of precipitation. However, the concentration of chemical ions in
precipitation may also be affected by many other factors in addition to below-cloud air
pollutant concentrations and in-cloud scavenging processes. For example, the
precipitation intensity may affect the scavenging efficiency of air pollutants below
cloud and hence influence wet deposition (Andronache, 2004b; Wang et al., 2014; Xu et
al., 2017;2019). Yuan et al. (2014) reported that in central North China high intensity
rainfall events of short duration (lasting less than 6 h) are dominant rather than long-
duration rainfall that is more common in the Yangtze River Valley. Therefore, the time
window for the definition of in cloud stage is very important for estimating the below
cloud and in cloud contributions. Previous studies have estimated the concentration of
chemical ions scavenged in-cloud based on the judgment that 5 mm of accumulated
precipitation is sufficient to identify the contribution of the in-cloud scavenging process
(Wang et al., 2009; Aikawa and Hiraki, 2009; Xu et al., 2017). Based on this approach,
the concentration of $NO_3^-$ and $SO_4^{2-}$ in cloud in Japan was found to be 0.70 and 1.30
mg/L, respectively (Aikawa and Hiraki, 2009). In Beijing, high concentration of $NH_4^+$,
$SO_4^{2-}$ and $NO_3^-$ during 2007 were found at 2.1~5.5, 3.1~14.9, 1.5~5.9 mg/L,
respectively (Wang et al., 2009;Xu et al., 2017).
In this study, a new method based on fitting a curve to the chemical ion
concentrations with successive rainfall increments has been developed to estimate the
contribution of the in-cloud process. As shown in Figure 1, an exponential curve is
fitted to the median, 25[th] and 75[th] percentiles of the chemical ion concentrations in each
fraction through the rainfall increments. Noted that, the fitted exponential curve is
applied to the combination of all 69 full events to estimate the yearly median
concentration of chemical ions in-cloud and to compare with the results from previously
reported method (i.e., median concentration after 5 mm increments). Besides, the
exponential approach to each unique event was also employed. Ideally, the
concentration of chemical ions stabilize at higher rainfall increments and this represents
the concentration in cloud. However, the decrease during each rainfall event is distinctly
different, and this regression method is not fully applicable to all rainfall events in
practice. Therefore, the exponential regression method is used to estimate the in-cloud
concentration under most circumstances, but where the decreasing trend with the
increment of rainfall is not significant, the average value of rainfall increments 6-8 of
the event is used. The below cloud contributions to wet deposition of each species are
then calculated using the following equations (1-2):

$$\text{Wetdep}_{below-cloud} = \sum_{i=1}^{n}(C_i - \overline{C}) \times P_i \qquad (1)$$

$$\text{Contribution}_{below-cloud} = \frac{\text{Wetdep}_{below-cloud}}{\sum_{i=1}^{n} C_i \times P_i} \qquad (2)$$

Where, $C_i$, and $\overline{C}$ represent the concentration of each chemical ion in fraction *i* and in
cloud and $P_i$ represents the volume of rainfall, while n represent the total fractions in
a rainfall event (equally to 8 in this study).
**3 Results and Discussion**
**3.1 Inter-annual variations in chemical components**
The Action Plan is launched in 2013 called "Ten rules" to improve the air quality in
China. It includes comprehensive control of industrial emission, non-point emission,
fugitive dust, vehicles, etc. It is also aimed to adjust and optimize the industrial
structure and promote economic transformation and upgrading, such as increase the
supply of clean energy. These actions are ensured to work by both of legislation and
market mechanism. According to the *Beijing Environmental Statement* published by the
Beijing Municipal Environmental Protection Bureau from 2013 to 2017, many
measures have been implemented to meet the Action Plan, including replacement
residential coal with electricity and natural gas, upgrading the emission standards of
gasoline, diesel vehicles and power plants, closing the high-emission enterprises.
Significant declines in atmospheric $PM_{2.5}$ concentration have been observed nationwide
between 2013 and 2017 during the Action Plan (Zhang et al., 2019). However, few
studies have investigated the benefits of the Action Plan for wet deposition. A
significant increase in $NO_3^-$ in precipitation of 7.6% was observed at a regional
background station in North China between 2003 and 2014 (Pu et al., 2017). A decrease
in the ratio of $SO_4^{2-}/NO_3^-$ mostly due to the decreasing of $SO_4^{2-}$ and increasing of $NO_3^-$
suggests the transformation of sulfuric acid type to a mixed type of sulfuric and nitric
acid in North China. However, the updated record especially after the Action Plan is
important to assess the mitigation of the air pollutants not only in the atmosphere but
also in rainfall. A nationwide investigation of the wet deposition of inorganic ions in
320 cities across China was recently made based on observations between 2011 and
2016 from the National Acid Deposition Monitoring Network (NADMN), which was
established by the China Meteorological Administration (Li et al., 2019). Briefly, both
$SO_4^{2-}$ and $NO_3^-$ across China experienced significant changes before and after 2014,
with increases from 2011 to 2014 and then decreases from 2014 to 2016.
In order to quantify the influence of the Action Plan on wet deposition in Beijing,
four years of observations of each rainfall event are considered in this study. Figure 2
shows the volume weighted average (VWA) of inter-annual mean concentrations of
$SO_4^{2-}$, $NO_3^-$, $NH_4^+$ and $Ca^{2+}$ observed in Beijing during 2014 to 2017 along with those
reported before 2010 from previous studies (Yang et al., 2012;Pan et al., 2012, 2013)
(more detail is provided in Table S1 in supplementary materials). A continuous
decrease in VWA concentrations between 1995 and 2017 is found for $SO_4^{2-}$, with
decreases of 3.1% $yr^{-1}$ in the earlier stage (1995-2010) and decreases of 9.8% $yr^{-1}$ in the
later stage (2014-2017). This is consistent with the annual changes in its emission and
concentration as shown in Figure 3, in which the emission and the concentration data
are collected from the yearly book of "*Environmental Bulletin in Beijing*" from 1994 to
2017. It is clearly shown the concentration of $SO_2$ experienced a sustainably decreasing
trend due to significant reduction of its emission from 1996 to 2017, with the decreases
rate is 4.5% $yr^{-1}$ and 13.9% $yr^{-1}$ in emission and 2.8% $yr^{-1}$ and 14.0% $yr^{-1}$ in
concentration during earlier stage and the later stage (the Action Plan period),
respectively. The significant declines in VWM concentration of $Ca^{2+}$ is found in
precipitation with the decreases rate as 36.1% yr-1 in 1995-2010 and 8.8% yr-1 in 2014-
2017. The emission and the concentration data of $Ca^{2+}$ are absent in this study. Instead,
the different of $PM_{10}$ and $PM_{2.5}$ ($PM_{10}$-$PM_{2.5}$) concentration during 2013-2017 have
been calculated to represent the coarse particles, which mainly contains the $Ca^{2+}$ as well.
The results show that the concentration decreased from 31.2 $\mu g/m^3$ in 2013-2014 to
24.0 $\mu g/m^3$ over 2015-2017. This indicates the improvement of coarse particles even
which is derived from crustal emissions have been made through the Action Plan. As
that is mentioned above, the Action Plan including emission reduction not only from
energy consumption of industry but also the fugitive dust in cities, which should result
the decline in $Ca^{2+}$. For $NO_3^-$ and $NH_4^+$, increases are found during the earlier stage
(~60%) and decreases in the later stage (12% for $NO_3^-$ and 25% for $NH_4^+$). As to $NO_x$
emission, the data in recent years have been collected. Although the clearly reduction
is found in the annual changes of emission from 2010, the ambient concentration of
$NO_2$ do not show a significant decreasing trend (~3.6% $yr^{-1}$) compared with $SO_2$ (14%
$yr^{-1}$). However, before the Action Plan, the decreasing ratio in concentration is only 1.8%
$yr^{-1}$, which is slower than the Action Plan period. Despite the increases of VWA $NO_3^-$
in precipitation during the earlier stage, the small decreases in later stage would also be
attributed to the Action Plan.
For a better understanding of the impacts of acidification on ecosystems, wet
deposition fluxes of the four major ions in precipitation are also plotted in Figure 2.
Similar variations are found as that presented in VWA of the four major ions.
Observations on S and N wet deposition (Pan et al., 2012; 2013) during 2007-2010
show the value of 21.5 kg S ha$^{-1}$ yr$^{-1}$ and 27.9 kg N ha$^{-1}$ yr$^{-1}$ (19.7 and 8.2 kg N ha$^{-1}$ yr$^{-1}$
$^1$ through $NO_3^-$ and $NH_4^+$) in Beijing, respectively. Compared with those results,
significant decreases (11.4 kg S ha$^{-1}$ yr$^{-1}$ and 23.6 kg N ha$^{-1}$ yr$^{-1}$) were observed in the
four-years measurements during 2014-2017 in this study. All four components in the
later stage show significant decreases, suggesting that the Action Plan, which was
implemented over this period, has a substantial impact. While $Ca^{2+}$ and $SO_4^{2-}$ played a
prominent role in precipitation during the earlier stage before 2010, $NH_4^+$ and $NO_3^-$
became the primary components in the later stage after 2010. It should be noted that
$NH_4^+$ has a double role in environment pollution because it mitigates acid rain through
neutralization, but also acidifies the soil by nitrification. Hence, while sulfur in
precipitation has been further reduced under the Action Plan, additional attention is
needed for nitrogen to prevent deterioration of the environment by acid rain resulting
from nitrate and ammonium.
**3.2 Relationship in concentrations in precipitation and the atmosphere**
Wet deposition of a substance involves its removal from the associated air mass. The
scavenging ratio H can be estimated by comparing the monthly average concentration
in precipitation with that in the air (Okita et al., 1996;Kasper-Giebl et al., 1999;Hicks,
2005;Yamagata et al., 2009). Xu et al. (Xu et al., 2017) first calculated the rainfall event
H based on the hourly concentration of aerosol components measured with an Aerodyne
Aerosol Chemical Speciation Monitor (ACSM) and AIM-IC in 2014. In this study four
years of observation of aerosol components have been undertaken by AIM-IC.
Measurements made in the 6 hours before each rainfall event are averaged to represent
the precondition of wet deposition precursors in the atmosphere. Figure 4 shows the
relationship between the major chemical ions in precipitation and in the air. The VWA
concentration of $SO_4^{2-}$, $NO_3^-$ and $NH_4^+$ (hereafter SNA) as well as $Ca^{2+}$ in each rainfall
event has been calculated and compared with that in the first 1 mm rainfall fraction,
F1#. As shown in Figure 4, positive correlations are found between the concentrations
of ions in precipitation and in air, with Pearson correlation coefficients (R) generally
higher than 0.7 (p<0.01). The concentration in the first fraction should represent a high
proportion of below-cloud scavenging due to the washout of air pollutants below clouds
by the first rainfall, while the VWA represents a greater contribution from in-cloud
removal (Aikawa and Hiraki, 2009;Wang et al., 2009;Xu et al., 2017). Thus, it is
reasonable that the correlations are stronger for the first fraction than for the VWA, see
Table 1. This indicates that the concentration of chemical ions in precipitation at the
start of rainfall is more greatly influenced by the air pollutants below the cloud. As
rainfall continues and below-cloud concentrations are reduced, there is an increased
contribution from in-cloud scavenging, which is less influenced by aerosols in the
surface layer. This is confirmed by the substantial difference in the two R coefficients
for the cation ion $Ca^{2+}$ (0.85 for the first fraction, 0.47 for the VWA), which often exists
in coarse particles below cloud. For the fine particle $SO_4^{2-}$ which is present both in and
below clouds (Xu et al., 2017), the difference in the two R coefficients is small. The R
coefficients for $NO_3^-$ and $NH_4^+$ show less difference than $Ca^{2+}$, but larger difference
than $SO_4^{2-}$. This may relate to their complicate sources from the ambient precursors.
For example, the $NO_3^-$ in precipitation is both from the fine and coarse particles (i.e.,
particulate $NO_3^-$) as well as the gaseous $HNO_3$, while the $NH_4^+$ in precipitation is mainly
from the fine particles in addition to $NH_3$.

326       The slope of the linear fits in Figure 4 can be used to calculate the scavenging ratio

$W$, which is the ratio of the ions concentration in precipitation (mg/L) and in air (μg/m³).
The $W$ ratio is $0.25 \times 10^6$, $0.16 \times 10^6$ and $0.15 \times 10^6$ for SNA, $SO_4^{2-}$, $NO_3^-$ and $NH_4^+$
respectively. This is similar to that reported for rainfall events in 2014 in Beijing
($0.26 \times 10^6$, $0.35 \times 10^6$ and $0.14 \times 10^6$ for SNA) by Xu et al. (2017) and consistent with
those estimated in the eastern United States ($0.11-0.38 \times 10^6$, $0.38-0.97 \times 10^6$ and $0.2-$
$0.75 \times 10^6$ for SNA) (Hicks, 2005). Compared with $SO_4^{2-}$ and $NH_4^+$, the scavenging ratio
for $NO_3^-$ shows larger differences between this study and previous studies,
corresponding to larger uncertainties to the R between the concentrations of ions in
precipitation and in air for VWA in Figure 4a (lower significance p<0.05). It should be
noted that the $W$ calculated in this study is based on the fine particles in air, which
may not represent the accurate reflection of the wet scavenging efficiency of SNA.
These uncertainties have been evaluated. For S, gas $SO_2$ was considered to testified its

role to the relationships. Figure S1 shows the relationships between the concentration of $SO_4^{2-}$ in precipitation and in air ($SO_4^{2-}$ in precipitation vs $SO_4^{2-}$, and $SO_4^{2-}$ in precipitation vs $SO_2+SO_4^{2-}$). The correlation coefficients R increased if the role of gas $SO_2$ was considered (R of $SO_4^{2-}$ in precipitation vs $SO_4^{2-}$ is 0.7, and R of $SO_4^{2-}$ in precipitation vs $SO_2+SO_4^{2-}$ is 0.75). However, the scavenging ratio $W$ was not changed, with the difference lower that 1%. For N, the contribution of gaseous $HNO_3$ to total inorganic nitrate is less than 2% in NCP according to Zhai et al. (2021), which can be ignored in this study. According to more than one-year measurements in Beijing (Tian et al., 2016), $SO_4^{2-}$, $NO_3^-$ and $NH_4^+$ in coarse particles account for 18%, 27% and 10%, respectively. The lower coefficient R in $NO_3^-$ than $SO_4^{2-}$ and $NH_4^+$ in Figure 4 is attributed to the absent of considering $NO_3^-$ in coarse particles. Besides, due to high concentration of $NH_3$ at ground surface over NCP (Pan et al., 2018), the $NH_4^+$ in precipitation from gaseous $NH_3$ cannot be ignored (Kasper-Giebl et al., 1999). The ratio of $NH_4^+/(2SO_4^{2-}+NO_3^-)$ in precipitation and in $PM_{2.5}$ was calculated. The lower ratio in precipitation than that in $PM_{2.5}$ was found, with 0.95-1.01 in precipitation and 1.35 in air. This implied the impacts of rich gas $NH_3$ at ground surface going into the precipitation by reacting with gaseous $HNO_3$ and forming as $NH_4NO_3$ after $(NH_4)_2SO_4$. Thus, the contribution of coarse particles and gases to the relationships of S and N compounds in precipitation and the atmosphere is not as important as the fine particles, except $NO_3^-$ in coarse particles and the gaseous $NH_3$, which should be considered in the future.

Wet deposition can affect much of the atmospheric column through in-cloud and below-cloud scavenging processes. The vertical column density (VCD) of $SO_2$ and $NO_2$ from satellite during 2000s to 2017 is used here to compare with the inter-annual variations in wet deposition in Beijing (Figure S2). Consistent variation of the VCD and the yearly VWA concentration in precipitation is found in S and N. A continuous decrease is found in VCD $SO_2$ from 2005 to 2017, matching the trend in $SO_4^{2-}$ deposition, while for VCD $NO_2$ shows an increase from 2001 to 2011, a decrease after 2011 and little change over the period 2014-2017. This implies that the Action Plan not only benefits air pollutants in the surface layer but also those in the total column. Due

to faster decreases in emissions of S than N (Zheng et al., 2018), the ratio of S/N in both
precipitation ($SO_4^{2-}/NO_3^-$, μeq/L) and air ($SO_4^{2-}/NO_3^-$, μg/m$^3$) are found to decrease,
with the change in ratio in precipitation at 17.5% yr$^{-1}$, 11% yr$^{-1}$ and 20.0 % yr$^{-1}$ during
1995-2010, 2014-2017 and 1995-2017, and in air at 12% yr$^{-1}$ during 2014-2017,
respectively, see Figure S3. This is also consistent with the trend reported in whole
China during 2000-2015 by Itahashi et al. (2018). The ratio of S/N in precipitation is a
useful index to investigate the relative contributions of these acidifying species. In
addition, the ratio of $NH_4^+/NO_3^-$ is investigated here and a clear decrease is found during
2014-2017 both in precipitation and in air. This indicates that $NH_4^+$ is decreasing faster
than $NO_3^-$. This evidence clearly confirms that nitrate should be the major target for air
pollution controls in the next Action Plan.
**3.3 Proportion of below cloud scavenging**
As described in section 2.3, the in-cloud ion concentration ($\overline{C}$, in Eq 1) can be derived
from the exponential fit of the observed rainwater concentrations. Table 2 lists the
asymptote value and the exponential fitting equation of the evolution of each ion
concentration in precipitation with the increment of rainfall. As shown, the asymptote
value (here after, exponential approach) based on the median data for $SO_4^{2-}$, $NO_3^-$ and
$NH_4^+$ was 3.18 mg/L, 2.32 mg/L and 1.39 mg/L, respectively. The $SO_4^{2-}$ and $NO_3^-$ are
within the range of reported in cloud concentrations for Beijing (3.33 mg/L and 2.75
mg/L for $SO_4^{2-}$ and $NO_3^-$ in Xu et al., 2017), while the $NH_4^+$ in this study is lower than
previous studies (2.51 mg/L in Xu et al., 2017 and 2.1-4.5 mg/L in Wang et al., 2009).
In-cloud concentrations for other ions, i.e., $Ca^{2+}$, $F^-$, $Cl^-$, $Na^+$, $K^+$ and $Mg^{2+}$, are 0.67
mg/L, 0.04 mg/L, 0.27 mg/L, 0.1 mg/L, 0.06 mg/L and 0.08 mg/L, respectively. For
comparison, the average concentration in fractions 6 to 8 (F6#~F8#) in each rainfall
event (here after, average approach) is used to estimate the in-cloud concentration for
events where successive rainwater concentrations do not show an obvious decrease or
where other factors such as precipitation intensity are important, see Table 2. Similar
results are found for most ions with the exponential and average approach except for
$NH_4^+$, $F^-$, $K^+$ and $Mg^{2+}$, where the maximum difference is less than 20% (Table 2). Thus,
the replacement of in-cloud concentration by the average value is acceptable for $SO_4^{2-}$,
$NO_3^-$, $Ca^{2+}$, $Cl^-$ and $Na^+$ but much uncertainty for the other ions. It is worth noting that
for all ions the average approach gives higher estimates of in-cloud concentrations, and
this can be recognized as an upper limit for in-cloud concentrations. It is also important
to note that the increased concentrations of ions in the latter fractions were observed in
few events in this study. This may due to the unique meteorological conditions and air
pollutants movement during each precipitation. Despite the longer precipitation
fractions in this study were collected, more longer fraction measurements and more
detailed analysis on the uncertainties are needed in the future. The influences of
meteorological conditions (i.e., rainfall type and intensity) are discussed in section 4.

408       The model study in Japan showed consistent fractions of in-cloud and below-cloud

scavenging to total wet deposition between simulated and observed values, except one
site, where is the region of high emission flux of $SO_2$. In this region, the simulated
below-cloud scavenging contribution was apparently greater than the observed results.
Specifically, the model shows the $SO_2$ and $HNO_3$ gases dominantly contributed to the
below-cloud scavenging of $SO_4^{2-}$ and $NO_3^-$ in the regions of high emission flux of $SO_2$,
in while the aerosol removal was dominated by the in-cloud scavenging process. In
their model set up, all of below-cloud gas $SO_2$ was assumed to be dissolved into
raindrop and be fully oxidized to $SO_4^{2-}$. However, as suggested by Seinfeld and Pandis
(2006), the aqueous equilibrium between ambient gas and precipitation cannot be
assumed due to the relatively short residence times of falling precipitation. Thus, the
assumptions used in Kajino et al. (2015) might overestimate the contribution of gas $SO_2$
to below-cloud scavenging. Besides, considering the large amounts of particles (60-90
$\mu g/m^3$ in mass concentration) below-cloud in Beijing, the gases compounds may be not
as important as that in simulation in Japan. According to the yearly book of
"*Environmental Bulletin in Beijing*" from 1994 to 2017, the yearly concentration of $SO_2$
has a dramatically decreasing from 26.5 $\mu g/m^3$ in 2013 to 8 $\mu g/m^3$ in 2017. This
relatively low-level concentration of $SO_2$ at surface may not contribute a dominant role
in wet deposition of $SO_4^{2-}$. Similar case in $NO_3^-$, the ratio of gas-phase $HNO_3$ and the
total $NO_3^-$ in the summer in Beijing is only 0.12 according to the measurement study of
Yue *et al*. (2013). The fraction of total inorganic nitrate as particulate nitrate instead of
gaseous nitric acid over the NCP increased from 90% in 2013 to 98% in 2017 (Zhai et
al., 2021), which means the gaseous nitric acid has been consumed by high level of
ammonia concentrations. We assumed the 10% ratio of gases added into the washout
process, which only leads to less 5% difference of below-cloud scavenging contribution
to total wet depositions. Anyway, for $NH_3$, there might be larger uncertainties, since the
high concentration of $NH_3$ at ground surface over NCP (Pan et al., 2018). Kasper-Giebl
et al. (1999) reported that 49-79% of $NH_4^+$ in precipitation are from particulate
ammonium, which indicate the large uncertainties of contribution from gases still exists
in the form of $NH_4^+$ wet deposition. The uncertainties are mainly from the indistinct
window for the in-cloud scavenging judgement due to high concentration of gas $NH_3$
at ground surface which is not easy to be scavenged completely during the short time
fraction measurements. This is also confirmed by the larger difference in below-cloud
contribution to $NH_4^+$ wet deposition than other ions estimated by the exponential
approach and the average approach in Table 2. As it mentioned above, more longer
fraction measurements as well as the influence of $NH_3$ to $NH_4^+$ wet deposition are
needed in the future.
Following Eq (2), the contribution of below-cloud scavenging to wet deposition in
each rainfall event during 2014-2017 are estimated from the in-cloud concentration.
Figure 5 shows the yearly VWA of SNA and $Ca^{2+}$ and the in-cloud and below-cloud
contributions. The ratio of below-cloud contribution to the four major components
based on the yearly median value of the in-cloud concentration is also shown in Figure
5. Benefiting from the Action Plan, the air quality at the surface layer have been
significantly improved (Zhang et al., 2019), which in turn leading to the decreases of
the below-cloud scavenging. In this study, it also shows the below-cloud contributions
of $SO_4^{2-}$, $NO_3^-$, $NH_4^+$ and $Ca^{2+}$ decreases from >50% in 2014 to ~40% in 2017. In 2017,
the contribution of below-cloud scavenging declines to lower than 40% for $SO_4^{2-}$ and
$NH_4^+$, but remains at 44% for $NO_3^-$. Over the four-year period 2014-2017, the average
yearly wet deposition for all ions and the below-cloud wet scavenging contributions are
given in Table 2. Similar to the concentrations in precipitation, the wet deposition of
$SO_4^{2-}$, $NO_3^-$, $NH_4^+$ decreased from 21.5 kgS ha$^{-1}$ yr$^{-1}$, 8.9 and 19.0 kg N ha$^{-1}$ yr$^{-1}$ during

2007-2010 (Pan et al., 2012; 2013) to 11.4 kgS ha$^{-1}$ yr$^{-1}$ (3.42×10$^3$ mg m$^{-2}$ yr$^{-1}$), 6.9 and

16.7 kgN ha$^{-1}$ yr$^{-1}$ (3.05×10$^3$ and 2.15×10$^3$ mg m$^{-2}$ yr$^{-1}$) during 2014-2017, respectively.

Below-cloud scavenging contributed to almost half of total deposition estimated with

the exponential approach (50~60%), higher than the average approach (40~50%).

**4 Factors influencing below-cloud scavenging**

Each precipitation event is unique in terms rainfall intensity, droplet sizes and

distribution, rainfall type (thunderstorms or deep convective scavenging), air

concentrations of chemical components, etc. The unique characterization of each

precipitation event was considered in calculation of the proportions from in-cloud and

below-cloud processes, as the exponential approach to each unique event was made.

The below-cloud proportions varied from 20% to 80% among the 69 rainfall events.

The influence of these factors affecting wet scavenging were investigated through the

correlation analysis between below-cloud proportions with the rainfall type as well as

the rainfall intensity.

**4.1 Rainfall type**

The rainfall over the North China Plain in summer time is usually determined by the

synoptic system such as the upper-level trough or the cold vortex. The 69 rainfall events

have been classified based on synoptic system according to records from the Beijing

Meteorological Service (http://bj.cma.gov.cn) with 33 events associated with upper-

level troughs, 23 events associated with a cold vortex and 13 events associated with

other systems. Figure 6 shows the contributions of below-cloud scavenging for the two

major systems. A high contribution from below-cloud scavenging is found for rainfall

events associated with an upper-level trough with the median contributions for $SO_4^{2-}$,

$NO_3^-$, $NH_4^+$ and $Ca^{2+}$ of 56.2%, 62.1%, 56.3% and 61.9%, respectively. In the contrast,

the contributions during rainfall events under cold vortex conditions are significant

lower, with the values of 42.2%, 44.5%, 41.7% and 53.9%, respectively. Rainfall events

associated with an upper-level trough are usually accompanied by orographic or frontal

precipitation and are characterized by long and continuous precipitation (Shou et al.,

2000). This suggests that below-cloud scavenging of chemical components is important

for this rainfall type due to air mass transport from outside Beijing. In contrast, rainfall

events associated with a cold vortex usually originate from strong thermal convection
and are characterized by short heavy rainfall (Zhang et al., 2008;Liu et al., 2016;Zheng
et al., 2020). This is common during the summer months in Beijing with deep
convective clouds (Yu et al., 2011;Gao and He, 2013), and suggests that there is a large
contribution from in-cloud scavenging to the total wet deposition.
**4.2 Precipitation intensity and rainfall volume**
To illustrate the impacts of rainfall on below-cloud aerosol scavenging, the relationship
between the below-cloud fraction and the rainfall volume and precipitation intensity are
investigated, see Figure 7. Negative correlations in below cloud fraction are found for
both the rainfall volume and precipitation intensity, although the relationship with the
former is stronger (R: 0.63~0.93 vs. 0.03~0.64). This is consistent with results for 2014
in Beijing reported by Xu et al. (2017). Atmospheric particles are efficiently removed
below cloud by washout at the beginning of precipitation events (almost 70% of SNA
is removed in the first 2-3 fractions, as shown in Figure 1). As the rainfall progresses,
in-cloud scavenging makes an increasingly important contribution as below-cloud
aerosol concentrations fall. Xu et al. (2017) found that heavy summertime rainfall
events with more than 40 mm of rainfall usually occur over very short periods of time,
usually 2-3 h. This heavy rainfall leads to the scavenging of aerosols in a relatively
localized region and prevents the compensation associated with transport of air
pollutants from outside the region during longer-duration light rainfall events. This
contributes to the decreased contribution of below-cloud scavenging during the high
intensity rainfall events.
**5 Conclusions**
This paper presents an analysis of below-cloud scavenging from four years of
sequential sampling of rainfall events in Beijing from May of 2014 to November of
2017. The concentration of ions in precipitation varied dramatically, with yearly volume
weighted averaged concentrations of $SO_4^{2-}$, $NO_3^-$, $NH_4^+$ and $Ca^{2+}$ decreasing by 39%,
12%, 25% and 35% between 2014 and 2017, respectively. Due to faster decreases in
$SO_4^{2-}$ than $NO_3^-$ both in precipitation and in the air during the observation period, there
is a significant decrease in S/N ratio in precipitation at 44% and in air at 48%.

Benefiting from the national Air Pollution Prevention and Control Action Plan, the sulfur has been further reduced, while the nitrogen, especially nitrate, needs further attention in the next Action Plan to prevent deterioration of the environment associated with acid rain and photochemical pollution.

A new method has been developed and employed to estimate the below-cloud contribution to wet deposition in Beijing. The new approach suggests that the contribution from below-cloud scavenging is greater than that estimated applying simpler approaches used in previous studies. Overall, the contribution of below-cloud scavenging to the wet deposition of the four major components is important at 50~60%. The contribution of below-cloud scavenging shows a decrease over the period 2014-2017 for $Ca^{2+}$, $SO_4^{2-}$ and $NH_4^+$, but little change for $NO_3^-$ during 2015-2017. Below-cloud scavenging also has a strong cleansing effect on air pollution, and the hourly concentration of $PM_{2.5}$ is found to decrease sharply as the rainfall events occur, even with the effects from wind swept out have been accounted for.

Rainfall types also influence the contribution of below-cloud scavenging. Seventy-five rainfall events during the four-year periods were classified based on the local synoptic conditions. Lower contributions from below-cloud scavenging (~40%) are found for the four major ions in rainfall events associated with a cold vortex, while higher contributions (~60%) occurred associated with an upper-level trough. Precipitation volume and intensity both show a negative correlation with the below-cloud fraction. This suggests that atmospheric particles are efficiently removed via below-cloud scavenging processes at the beginning of precipitation events. As the event progresses, rainfall in the later fractions shows a greater contribution from in-cloud scavenging processes as aerosols in the surface layer have already been removed. To better understand the mechanism of below-cloud scavenging processes, high resolution of measurement both in precipitation and in the air especially at the beginning of rainfall events are needed in the future.

**Data availability.**

To request observed data for scientific research purposes, please contact Baozhu Ge at
the Institute of Atmospheric Physics, Chinese Academy of Sciences, via email
(gebz@mail.iap.ac.cn).

**Author contribution**

BG and ZW designed the whole structure of this work, and prepared the manuscript
with contributions from all co-authors. DX, XY, JW and QT helped with the data
processing. OW, XC and XP was involved in the scientific interpretation and discussion.

**Competing interests**

The authors declare that they have no conflict of interest

**Acknowledgment**

We appreciate CNEMC for providing the data of the 6 criteria pollutants in Beijing. We
also appreciate Beijing Municipal Environmental Monitoring Center for providing the
aerosol components data in Beijing. This work is supported by the National Natural
Science Foundation of China (Grant No 41877313, 91744206, 41620104008), Priority
Research Program (XDA19040204) and the Key Deployment Program (ZDRW-CN-
2018-1-03) of the Chinese Academy of Sciences.

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

Table 1. Correlation of the concentrations of major ions in air in the six hours before
rainfall with those in precipitation. Pearson correlation coefficients are presented for
monthly volume weighted average (VWA) concentrations and for the first fraction (F1[#])
in each event.

|  | $SO_4^{2-}$ (n=13) | $NO_3^-$ (n=14) | $NH_4^+$ (n=13) | $Ca^{2+}$ (n=9) |
|---|---|---|---|---|
| VWA | 0.70[a] | 0.53[b] | 0.65[a] | 0.47 |
| F1[#] | 0.76[a] | 0.62[a] | 0.77[a] | 0.85[a] |

Note: "[a]" and "[b]" represent significant correlations at $p<0.01$ and $p<0.05$, respectively.

Table 2. Exponential fitting for the concentrations of major ions in different fractions of rainfall, and the contribution of below-cloud scavenging
to total deposition.

| Chemical components | Exponential Fitting for 50th percentile[a] | $R^2$ (n=11) | Asymptote value (mg/L) | Below cloud %[b] | Average of F6#-F8# (mg/L) | Below cloud %[c] | Difference %[d] | Total wet deposition (mg/m²/yr) |
|---|---|---|---|---|---|---|---|---|
| $SO_4^{2-}$ | $y=3.17+10.28\ e^{-0.51x}$ | 0.85 | 3.18 | 50% | 3.33 | 48% | <3% | 3423.3 |
| $NO_3^-$ | $y=2.32+11.03\ e^{-0.45x}$ | 0.81 | 2.32 | 59% | 2.59 | 54% | <6% | 3046.5 |
| $NH_4^+$ | $y=1.39+5.81\ e^{-0.28x}$ | 0.79 | 1.39 | 65% | 1.95 | 51% | <9% | 2149.5 |
| $Ca^{2+}$ | $y=0.67+6.81\ e^{-0.6x}$ | 0.93 | 0.67 | 52% | 0.72 | 48% | <6% | 746.0 |
| $F^-$ | $y=0.04+0.24\ e^{-0.34x}$ | 0.91 | 0.04 | 56% | 0.05 | 40% | <10% | 49.0 |
| $Cl^-$ | $y=0.27+2.2\ e^{-0.6x}$ | 0.95 | 0.27 | 53% | 0.29 | 50% | <5% | 309.7 |
| $Na^+$ | $y=0.1+1.34\ e^{-0.94x}$ | 0.91 | 0.10 | 64% | 0.10 | 64% | <1% | 150.6 |
| $K^+$ | $y=0.06+0.49\ e^{-0.47x}$ | 0.89 | 0.06 | 64% | 0.07 | 58% | <9% | 89.8 |
| $Mg^{2+}$ | $y=0.08+0.81\ e^{-0.4x}$ | 0.83 | 0.08 | 61% | 0.11 | 46% | <13% | 109.2 |

[a] fitting for the median of each fraction in different rainfall events; [b] below cloud portion calculated based on the fitting curve; [c] below cloud portion
calculated based on the average value of fractions 6 to 8 (F6#~F8#) in rainfall events; [d] difference in concentrations between adjacent 1 mm
increments after 5 mm accumulated precipitation.


**Figures and captions**

**Figure 1.** Concentrations of $SO_4^{2-}$ (a), $NO_3^-$ (b), $NH_4^+$ (c) and $Ca^{2+}$ (d) in each 1-mm fraction of rainfall (i.e., F1#, F2#, …) over different rainfall events in the observation periods. The red line shows an exponential fitting using the 50th percentile of the data and the red shading indicates the range between the 25th and 75th percentiles.

**Figure 2.** Time series of annual volume weighted average (VWA) concentration and wet deposition of the four major components $NH_4^+$ (a), $Ca^{2+}$ (b), $SO_4^{2-}$ (c) and $NO_3^-$ (d) in precipitation in Beijing.

**Figure 3.** Annual changes in emission and concentration of $SO_2$ and $NO_x$ in Beijing, data is collected from the yearly book of "*Environmental Bulletin in Beijing*" from 1994 to 2017.

**Figure 4.** Relationships between the concentration of $NO_3^-$ (a), $SO_4^{2-}$ (b), $NH_4^+$ (c) and $Ca^{2+}$ (d) in precipitation and in air in the 6 h before each precipitation event. The red square and blue triangle represented the relationships between the concentration of ions in air with that in F1# and in VWA, respectively.

**Figure 5.** The annual volume weighted average below-cloud and in-cloud portion of $SO_4^{2-}$ (a), $Ca^{2+}$ (b), $NO_3^-$ (c), and $NH_4^+$ (d) during 2014-2017. The ratio of annual median below-cloud contribution for each component is represented as the black line in each panel. The mark #M and #A in the ratio of below-cloud represent the estimation based on the median value and average value of in-cloud concentration in each year, while the first quartile and the third quartiles are also included in the figure.

**Figure 6.** Contribution of below-cloud scavenging during rainfall events associated with different synoptic conditions.

**Figure 7.** Contribution of below-cloud scavenging in events with different rainfall volume and precipitation intensity

836

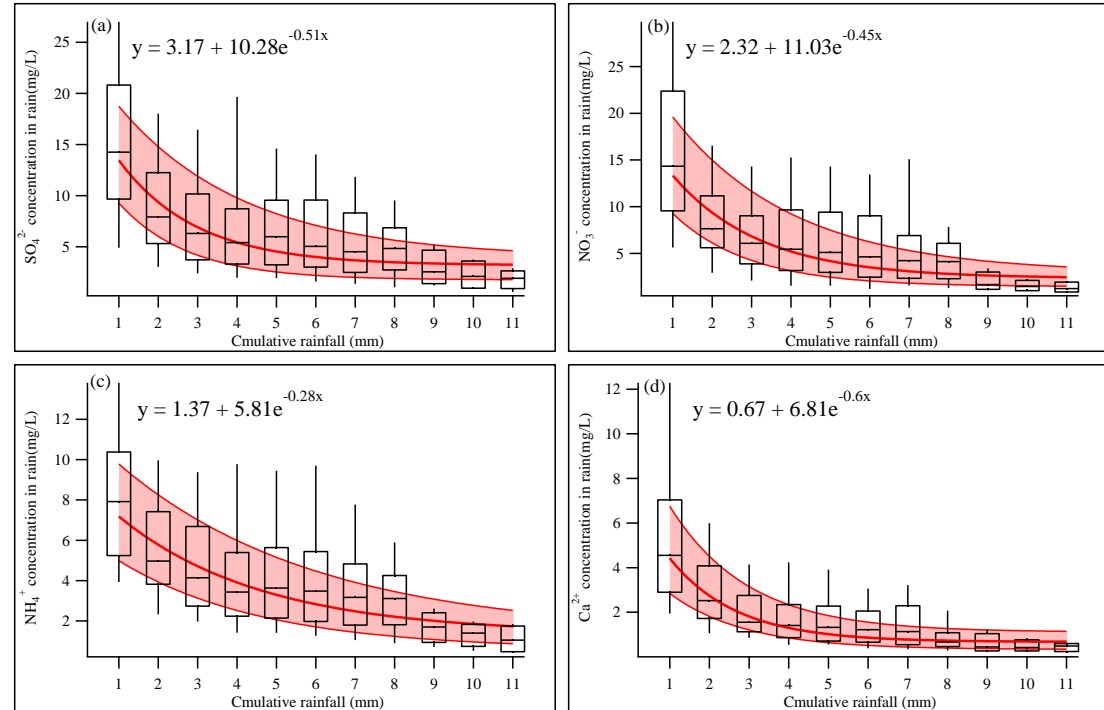

837

Figure 1. Concentrations of $SO_4^{2-}$ (a), $NO_3^-$ (b), $NH_4^+$ (c) and $Ca^{2+}$ (d) in each 1-mm
fraction of rainfall (i.e., F1#, F2#, …) over different rainfall events in the observation
periods. The red line shows an exponential fitting using the 50[th] percentile of the data
and the red shading indicates the range between the 25[th] and 75[th] percentiles.



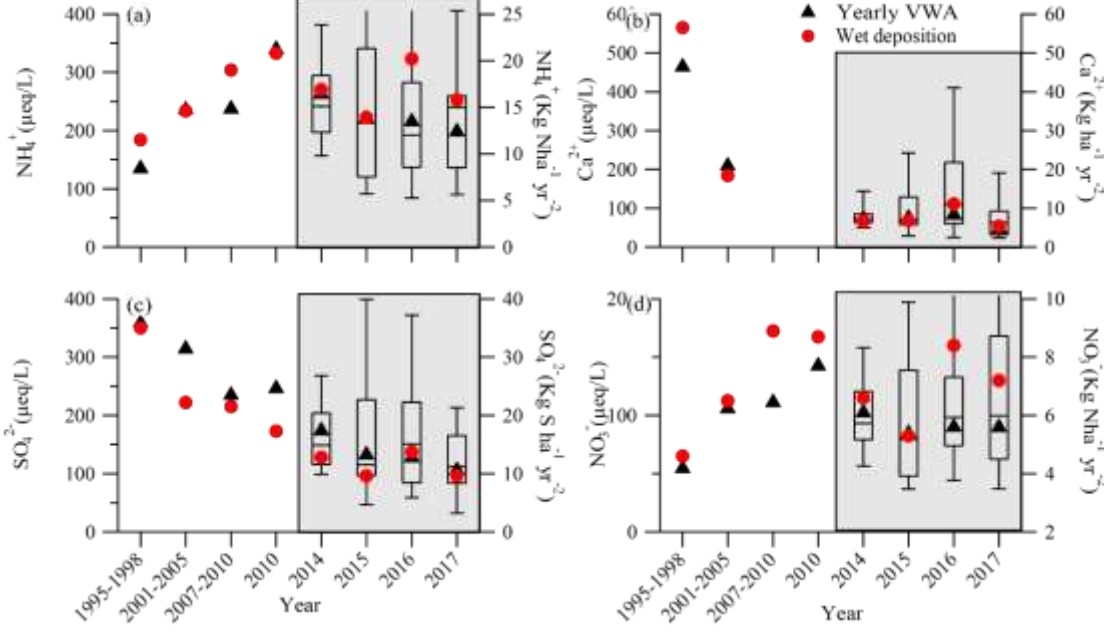

Figure 2. Time series of annual volume weighted average (VWA) concentration and
wet deposition of the four major components $NH_4^+$ (a), $Ca^{2+}$ (b), $SO_4^{2-}$ (c) and $NO_3^-$
(d) in precipitation in Beijing.

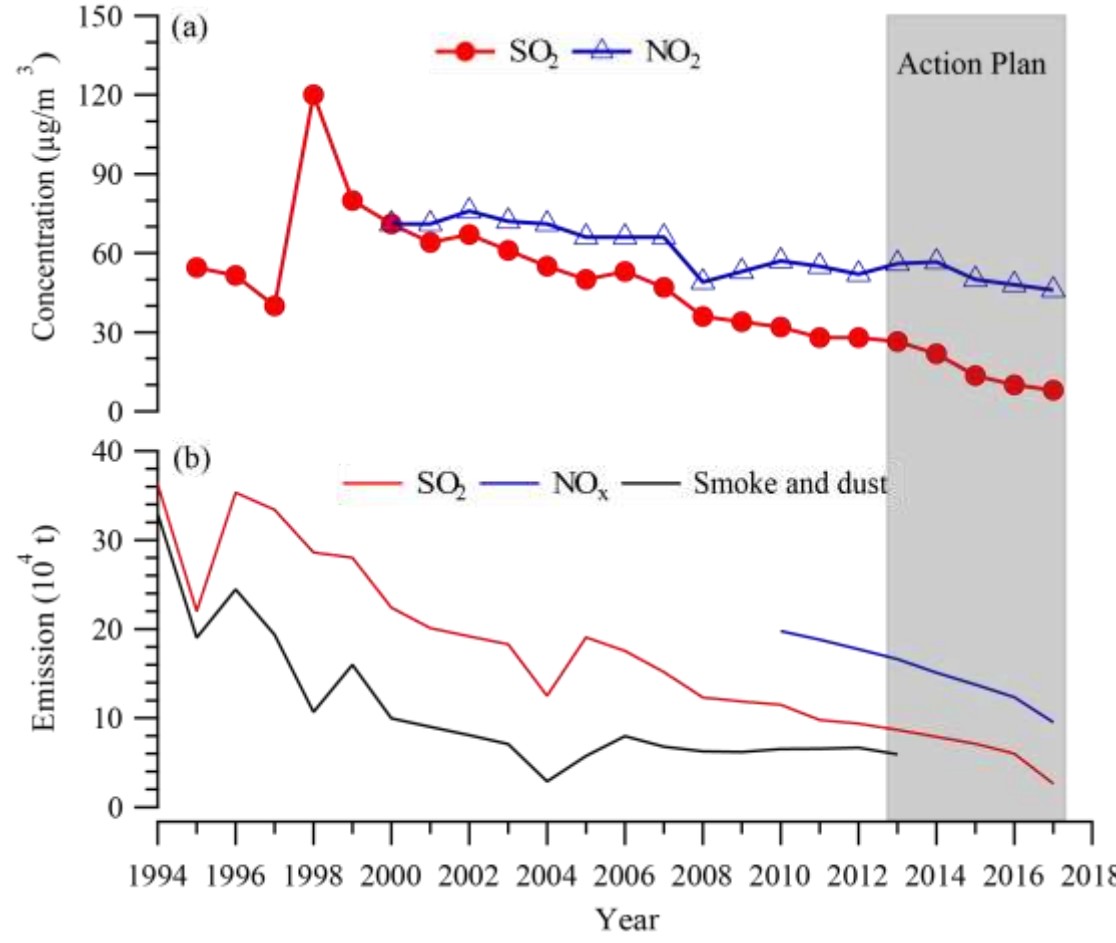


Figure 3. Annual changes in emission and concentration of $SO_2$ and $NO_x$ in Beijing,
data is collected from the yearly book of "*Environmental Bulletin in Beijing*" from
1994 to 2017.

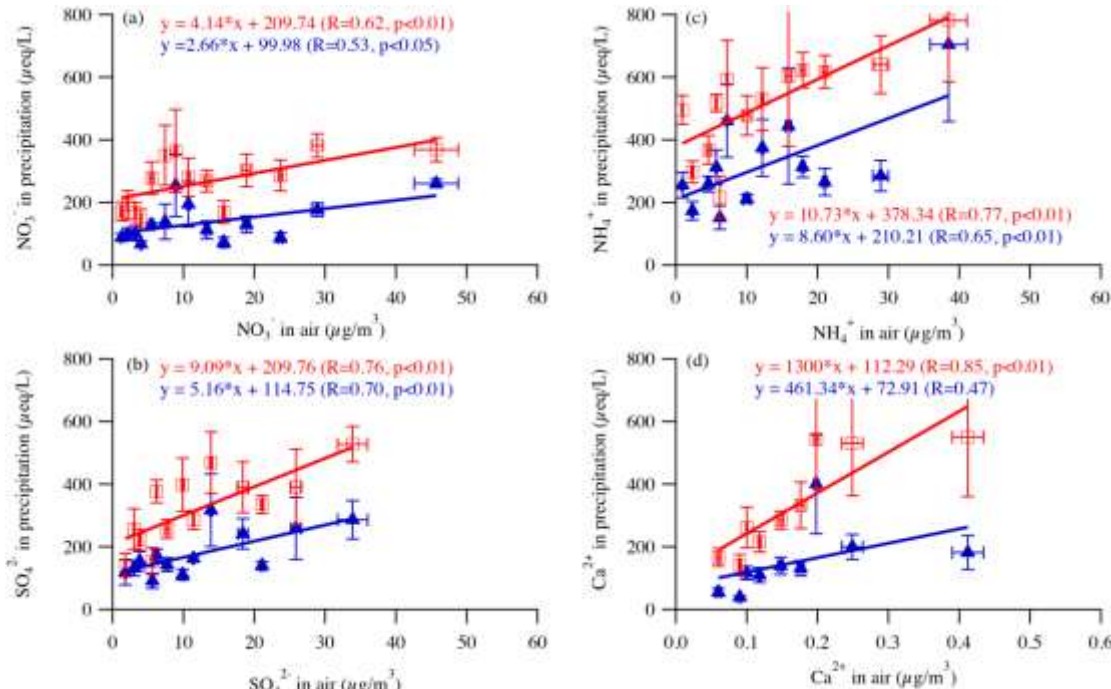

Figure 4. Relationships between the concentration of $NO_3^-$ (a), $SO_4^{2-}$ (b), $NH_4^+$ (c) and $Ca^{2+}$ (d) in precipitation and in air in the 6 h before each precipitation event. The red square and blue triangle represented the relationships between the concentration of ions in air with that in F1# and in VWA, respectively.

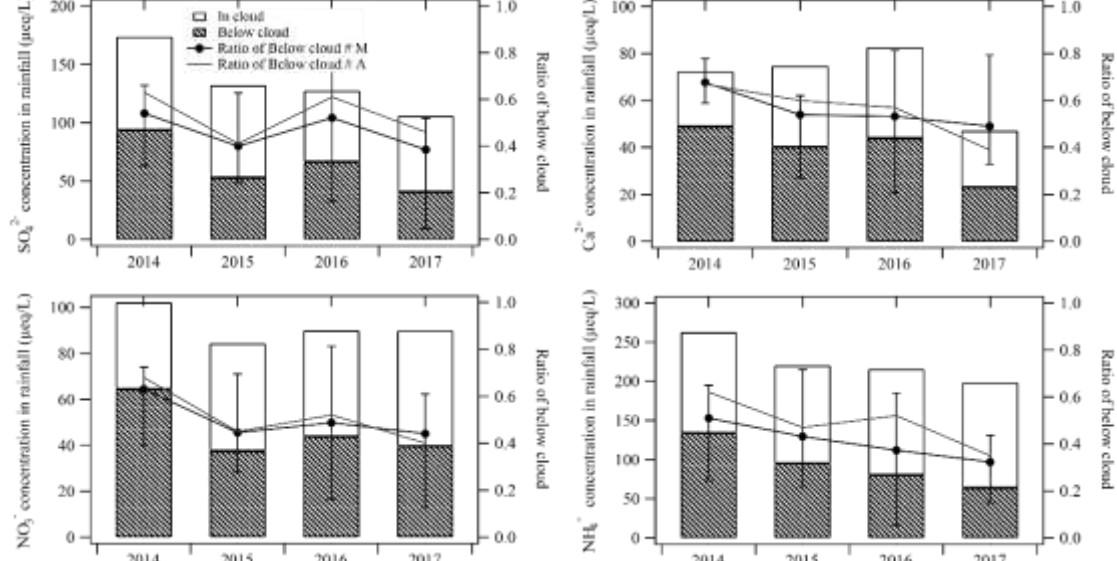

Figure 5. The annual volume weighted average below-cloud and in-cloud portion of
SO$_4^{2-}$ (a), Ca$^{2+}$ (b), NO$_3^-$ (c), and NH$_4^+$ (d) during 2014-2017. The ratio of annual
median below-cloud contribution for each component is represented as the black line
in each panel. The mark #M and #A in the ratio of below-cloud represent the estimation
based on the median value and average value of in-cloud concentration in each year,
while the first quartile and the third quartiles are also included in the figure.

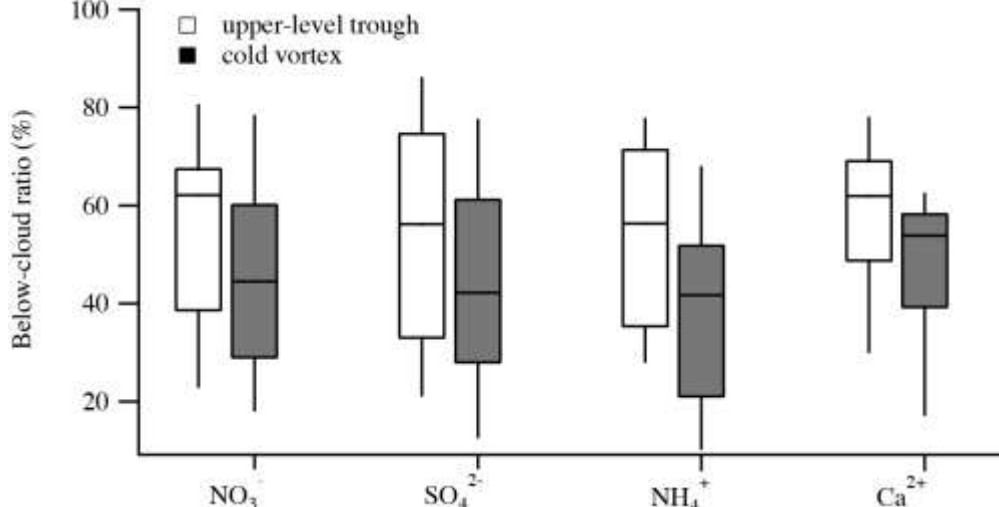

Figure 6. Contribution of below-cloud scavenging during rainfall events associated
with different synoptic conditions.

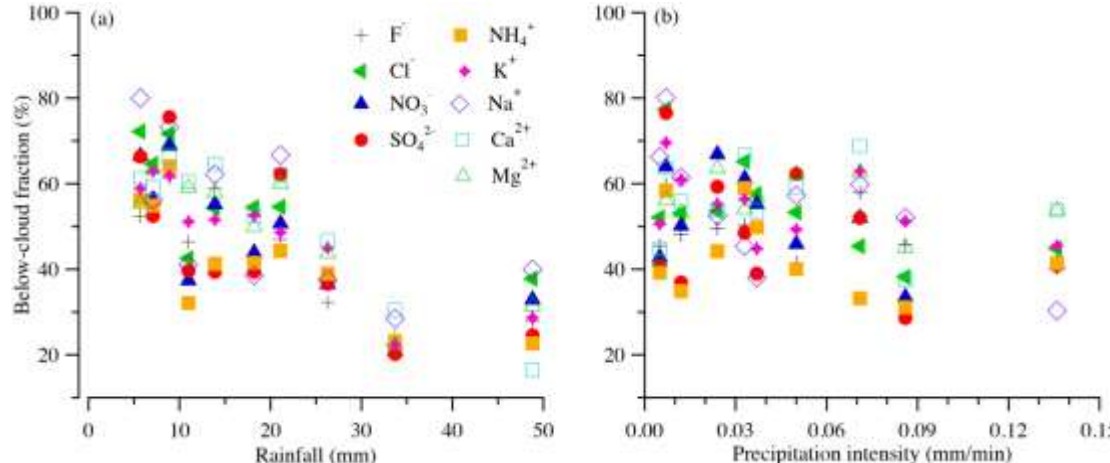

Figure 7. Contribution of below-cloud scavenging in events with different rainfall
volume and precipitation intensity.