# Peer review of "Inter-annual variations of wet deposition in Beijing during 2014-2017"

_Atmospheric Chemistry and Physics, 2020_

## Referee Comment (RC1) · Anonymous Referee #1 · 25 Dec 2020

The manuscript entitled 'Inter-annual variations of wet deposition in Beijing during 2014-2017: implications of below-cloud scavenging of inorganic aerosols' written by Baozhu Ge investigated the long-term variation of wet deposition at Beijing site during 2014-2017, The topic is interesting and provides important results for wet deposition process. However, before the considered publication from ACP journal, I would like to suggest to address the following concerns.

Major points:

1. From L109, the total of 69 full events and 6 extended events were recorded during the sampling period from 2014 to 2017. I might miss the description, but what are

the available numbers at each year? From the limited observation number, it could be doubtful the long-term trends described in Section 3.1. From conclusion section, I found that the exact time period is May 2014 to November 2017. In this sense, the data on 2014 might be different because the winter and early-spring season observation is not included in this year. How can we consider this point for long-term behavior?

2. It is ambiguous that what satellite data is used here only from the description in L256-258 (and related supplement). In addition, satellite observed pixel will be only one (or a few) to correspond Beijing. Is it appropriate to use such limited data? To clarify the data usage, the detail is needed at least in supplemental material.

Minor points:

1. L65: Is "CMAQ" widely known as benchmark model? This model is used without any explanations before.

2. L133: Correct to use subscript for "4" in "NH4+".

3. L241-244: Need discussion for NO3- and NH4+.

4. L249-251: Does this imply that the scavenging ratio itself would be constant over the world even though the air pollution level is different?

5. It will be better to unify the wording of "washout/rainout" or "below-cloud/in-cloud scavenging" throughout manuscript.

---

## Referee Comment (RC2) · Anonymous Referee #2 · 29 Dec 2020

This study analyzed concentrations of sulfate, ammonium, nitrate and other inorganic ions in precipitation and fine PM and estimated the fraction of the wet deposition attributed to in-cloud and below-cloud scavenging processes. The sequential sampling of precipitation over the course of a precipitation event provides interesting insight into changes in the precipitation chemistry that cannot be obtained from daily or weekly precipitation monitoring that are reported in many studies. The study introduced a newer approach to estimate the in-cloud and below-cloud wet scavenging proportions. In spite of the novel aspects in this study, the scientific discussions on wet scavenging of acidifying pollutants needs to be improved.

[Figure]

The results and discussion center on the wet scavenging of inorganic ions in PM2.5. There was hardly any discussion on the wet scavenging of coarse particles and gas-phase S and N compounds that can also contribute substantially to wet deposition of sulfate, nitrate and ammonium. The paper discussed relationships between precipitation and ambient air concentrations of inorganic ions; however, the latter was based on components in PM2.5. How would the relationship change if coarse PM and gas-phase compounds were included? Similarly for the below-cloud scavenging results, there needs to be a more balanced discussion on particle and gas-phase scavenging.

A more detailed explanation as to why the beginning and latter precipitation fractions represent below-cloud and in-cloud scavenging, respectively, is necessary since most of the results from this study are based on this key assumption. This data analysis method seems to assume that below-cloud and in-cloud scavenging have to occur in sequence.

The discussion on temporal variations in precipitation concentration of inorganic ions is lacking some depth. Further analysis with annual emissions data are necessary to explain interannual changes and appreciate the impact of the air pollution control policies implemented.

The way the below-cloud percentages are reported in the paper raises some critical questions. In this version, only the median below-cloud percentage were reported. It is uncertain how representative are the median values considering that each precipitation event is unique in terms of rainfall intensity, precipitation form (rain, snow, or sleet), air concentrations of chemical components, type of weather system, etc. It is important to state or discuss under what conditions apply to those below-cloud percentages and their limitations. A range of the below-cloud percentages should also be included as well as some discussion on the variability of the below-cloud scavenging percentages.

Specific comments: L22-24: Why would the Action Plan result in declines in Ca2+ given that it is mainly derived from crustal emissions?

L25: "An improved sequential sampling method. . ." This is vague. The improvement to the method needs to be clarified in the abstract. It is not the improvement to the sequential sampling method that is described in the paper, but rather the improvement to the method of estimating the below-cloud scavenging proportion.

L27: Suggested revision, "below-cloud scavenging accounts for one half to two thirds of wet deposition". To be more concise, only the second part of the sentence is necessary because it is very similar to the first part of the sentence.

L30: Was there a continuous year-to-year decline in the percentages, i.e. in 2015 and 2016? If so, this should be stated. If not, the next sentence, which suggests the result is due to the Action Plan, should not be included in the abstract because there is no clear decreasing trend.

L36-37: ". . .clearly identifies oxidized nitrogen species as a major target for future air pollution controls." This statement seems to be based only on the lack of declining trends in airborne and precipitation nitrate. The observed concentrations of nitrate also need to be considered compared to other N or S compounds. In precipitation, it seems that the VWA of nitrate was similar to that of sulfate and ammonium was even higher than nitrate. It is not clear why the policy recommendation is to target oxidized nitrogen and not other N or S compounds as well.

L44-46: Do these factors affect below-cloud rain scavenging only or both rain and snow?

L57-58: For which chemical components do these results apply to? Can you provide details on the modeled and measured scavenging coefficients?

L60-64: Here you argue that uncertainties in below-cloud scavenging coefficients is a potential reason for model underestimation of nitrate and sulfate wet deposition. How important are other sources of model uncertainties, such as N and S emissions, chemical transformation, changes in other ambient N and S compounds, etc.?

[Figure]

L78-80: "The chemical components in later increments of rainfall are thought to be less influenced by the below-cloud scavenging process than by the in-cloud rainout process". The wording is confusing. Why does the latter precipitation increments in sequential sampling represent in-cloud scavenging, while the start of the precipitation increments represent below-cloud scavenging? At what point during a precipitation event does the dominant wet scavenging process change from below-cloud to in-cloud? Is this transition point the same for all precipitation events and locations? Each precipitation event is unique in terms rainfall intensity, droplet sizes and distribution, precipitation form (rain, snow, or sleet), air concentrations of chemical components, etc. Meteorology varies with location; some locations are frequently affected by thunderstorms or deep convective scavenging. Can you address whether these factors are taken into consideration when using the sequential precipitation sampling method to estimate in-cloud or below-cloud scavenging contributions?

L105: How much precipitation is collected in the eighth fraction? These details should be included in the paper.

L109: In the 69 full events, what were the total precipitation amounts collected? Are they roughly the same for each event or do they vary greatly? At what point do you have to conduct manual sampling to collect the remainder of the precipitation? These details should be included in the paper.

L141-147: Is there additional evidence that can be presented to justify the assumption that precipitation at the start of an event represents below-cloud scavenging while latter precipitation represents in-cloud scavenging? Some studies have measured concentrations of chemical components in cloud water as an indication of in-cloud scavenging; however, the method in this study are based solely on the precipitation at the surface. One initial criticism I have on this assumption is why below-cloud and in-cloud processes have to occur one after another instead of simultaneously. A full justification is necessary given that the major results of this study depend on this analysis and interpretation.

[Figure]

L157: In previous studies, it seems that the 5 mm accumulated precipitation cut-off was based on the point where there was a lack of change in the precipitation concentrations. This is not entirely subjective.

L164-174: Clarity on the methodology is needed. You mentioned that each precipitation event is unique in terms of the decreasing rate in the precipitation concentrations. What is the reason for combining all the precipitation events prior to fitting a regression curve? Why not fit a curve for each event? What is the purpose of fitting the data through the first and third quartiles – can this serve as an estimate of the uncertainties? "In theory, the concentration of chemical ions stabilize at higher rainfall increments and this represents the concentration in cloud." A brief explanation of the theory should be provided in the paper.

Equations 1,2: What does n represent? n in eq. 1 is different from the n in eq. 2.

L185: Some background information on the Action Plan is needed in the introduction to understand what effects these policies may have on air quality. Which pollutants does the Action Plan target and what was implemented?

Section 3.1: Why are the results of other inorganic ions not discussed? The annual changes in the VWM concentrations should be discussed with changes in emissions of the precursor pollutants, such as SOx and NOx emissions, and policies that were introduced to manage those emissions. There should also be more discussion on what was causing the significant declines in sulfate and Ca2+ in the earlier period. Air pollution control measures typically target major anthropogenic contaminants, but here you have shown significant decreases in Ca2+ which is largely from soil emissions. It is not clear how the Action Plan played a role in the reduction of VWM Ca2+. For a better understanding of the impacts of acidification on ecosystems, wet deposition fluxes should be presented as well. While there may be significant decreases in VWM concentrations, interannual variability in precipitation amounts may result in less significant decreases in wet deposition fluxes.

L219: H usually denotes a Henry's Law constant. It is more suitable to use W or SR to denote scavenging ratios as this is consistent with literature.

L234: "while the VWA represents a greater contribution from in-cloud removal…" It was stated earlier that latter fractions of the accumulated precipitation represents in-cloud removal. Why was the VWA concentrations (which may include below-cloud and in-cloud contributions) used instead of the concentrations in the latter precipitation fractions?

L237-238: "This indicates that the concentration of chemical ions in precipitation at the start of rainfall is more greatly influenced by aerosols below the cloud." This conclusion is not correct because the scavenging of acidic gases like HNO3 and SO2 can also contribute to nitrate and sulfate in precipitation. Another issue with the correlation analysis between precipitation and air concentrations (in Fig. 3) is that only fine aerosols were considered. Would there be any changes to the correlation results if acidic gases and coarse aerosols were included?

L245: The scavenging ratios of SNA do not include coarse particles since only PM2.5 chemical composition were measured. It also does not include the wet scavenging of acidic gases. This needs to be mentioned in the paper because typically scavenging ratios take into account all the chemical species in air that can undergo wet deposition. It seems that the scavenging ratios of SNA calculated in this study is not an accurate reflection of the wet scavenging efficiency of SNA.

L264-266: The percentages should be converted to annualized percentages. Some of the time periods are much longer than others, which makes it appear that the rate of decrease is larger.

L269: "The chemical composition of precipitation is directly related to the amount of precipitation, and as a consequence it is difficult to identify inter-annual variations in chemical concentrations." This statement is incorrect. There are other factors affecting the precipitation chemistry besides the precipitation amount. Section 3.3: An exponential curve was used to fit the data points and then the horizontal asymptote was used to determine the precipitation concentration representative of rainout. One issue that I have with this approach is that there were only 11 data points or precipitation fractions. Previous studies (e.g. Aikawa and Hiraki, 2009) that conducted sequential sampling of precipitation found that the precipitation concentrations could increase in the much latter fractions of precipitation. It seems that more precipitation fractions need to be collected in order to observe what happens to the precipitation concentrations.

L303-308: "Benefiting from the Action Plan, the below-cloud contributions..." This part may not be necessary because it was discussed earlier that the policies in the Action Plan help reduced the VWM of inorganic ions. The Action Plan did not directly affect the below-cloud scavenging process. Rather, it improved the air quality which in turn decreased the VWM concentrations.

Section 3.4: I think this section is not necessary because the findings are not substantially new from what has already been discussed in the paper. Here you are showing the effects of rainfall on ambient air sulfate. The washout effects were already discussed in Figure 1, which shows the decrease in VWM concentrations of the inorganic ions with increasing rainfall.

L392: "highest contribution for NH4+ at 65% and lowest for SO42- at 50%". This was stated in the conclusion and abstract, but was not elaborated in the results. I expected the highest contribution to below-cloud wet scavenging would have been from Ca2+ or Na+ given that they are predominantly associated with coarse particles and locally emitted.

---

## Author Comment (AC1) · 6 Mar 2021

The authors appreciate the reviewers for reviewing our manuscript and providing constructive comments. As suggested, we carefully revised the manuscript thoroughly according to the valuable advices, as well as the typographical, grammatical, and bibliographical errors. Listed below are our point-by-point responses in blue to the review's comments (in italic).

**Anonymous Referee #1**

*The manuscript entitled 'Inter-annual variations of wet deposition in Beijing during 2014-2017: implications of below-cloud scavenging of inorganic aerosols' written by Baozhu Ge investigated the long-term variation of wet deposition at Beijing site during 2014-2017, The topic is interesting and provides important results for wet deposition process. However, before the considered publication from ACP journal, I would like to suggest to address the following concerns.*

**[Response]:** We thank the reviewer for the valuable comments. We have prepared the point-by-point responses to address the reviewer's comments as shown below.

*Major points:*
*1. From L109, the total of 69 full events and 6 extended events were recorded during the sampling period from 2014 to 2017. I might miss the description, but what are the available numbers at each year? From the limited observation number, it could be doubtful the long-term trends described in Section 3.1. From conclusion section, I found that the exact time period is May 2014 to November 2017. In this sense, the data on 2014 might be different because the winter and early-spring season observation is not included in this year. How can we consider this point for long-term behavior?*

**[Response]:** Thanks for the comments. The available numbers of full events at each year are 15, 16, 20 and 18, respectively. During 2014-2017, a total of 104 precipitation events, which is almost 690 precipitation samples, were collected. Of the total number of precipitation events, 33 events (32%) were discarded from the sequential sampling analysis due to low rainfall amounts (<8 mm), which cannot satisfy the full events. Note that the precipitation samples are only rainfall (excluded snow). Most of rainfalls were occurring summer and only 1-2 events were during the winter and early-spring season in Beijing. Thus, the time period which is started from May 2014 would not lead to much difference in 2014 from the other years. This is also reflected from the similar full events at each year. Besides, the results before 2014 from the previous studies in Beijing were collected to compare with our data during 2014-2017 for the purpose of describing the long-term trends variations. We respected to the reviewer's comments,

the limited data cannot fully reflect the long-term trends of precipitation chemistry. The descriptions of "long-term trends" were changed to "inter-annual variations" in the whole text. The detailed descriptions on the rainfall events collected and selected in this study were also added in section 2.1 as "*During 2014-2017, a total of 104 precipitation events, which is almost 690 precipitation samples, were collected. Of the total number of precipitation events, 33 events (32%) were discarded from the sequential sampling analysis due to low rainfall amounts (<8 mm), which cannot satisfy the rules of full events. Altogether, 69 full events and 6 extended events were recorded over the 2014-2017 period in Beijing, as 15, 16, 20 and 18 events at each year, respectively.*"

*2. It is ambiguous that what satellite data is used here only from the description in L256-258 (and related supplement). In addition, satellite observed pixel will be only one (or a few) to correspond Beijing. Is it appropriate to use such limited data? To clarify the data usage, the detail is needed at least in supplemental material.*

**[Response]:** The level 3 product of the ozone monitoring instrument (OMI) satellite data were used in this study. The OMI instrument, which is board on the Aura satellite, can measures the solar radiation backscattered by the atmosphere and surface in the Earth (Torres et al., 2002). The data is stored in the HDF-EOS format with a resolution of 0.25×0.25, which covers the total vertical column density for $SO_2$ and $NO_2$, the standard errors, cloud information, data quality flags, and the latitude/longitude information. The OMI VCD $SO_2$ and $NO_2$ data were derived by the algorithm of a principal component analysis (Li et al., 2013), and were widely used in local regions such as Henan province (Zhang et al. 2017) and the major cities (including Beijing) in China (Tang et al. 2019). There are almost 25 pixels covering the whole domain of Beijing. To compare with the yearly trends of sulfur and nitrogen in precipitation, the vertical column density data observed from the space is better than that only observed at the surface layer. Detailed description of OMI data has been added in the supplemental material.

*Minor points:*
*1. L65: Is "CMAQ" widely known as benchmark model? This model is used without any explanations before.*

**[Response]:** The CMAQ model is Community Multiscale Air Quality model and is

added in the revised manuscript.

*2. L133: Correct to use subscript for "4" in "NH4+".*

**[Response]:** Thanks for the correction. The subscript for "4" in $NH_4^+$ has been corrected in the revised manuscript.

*3. L241-244: Need discussion for NO3- and NH4+.*

**[Response]:** Thank for your suggestion. The discussion on the ions are also included in the revised manuscript, which is as: *The R coefficients for $NO_3^-$ and $NH_4^+$ show less difference than $Ca^{2+}$, but larger difference than $SO_4^{2-}$. This may relate to their complicate sources from the ambient precursors. For example, the $NO_3^-$ in precipitation is both from the fine and coarse particles (i.e., particulate $NO_3^-$) as well as the gaseous $HNO_3$, while the $NH_4^+$ in precipitation is mainly from the fine particles in addition to $NH_3$.*

*4. L249-251: Does this imply that the scavenging ratio itself would be constant over the world even though the air pollution level is different?*

**[Response]:** No. The scavenging ratio is not a constant value over the world. It should be different due to different air pollution level as well as different rainfall type. The scavenging ratio represents the scavenging efficient of each air pollutant that is removed from the atmosphere by rainfalls. The statements in the revised manuscript are changed as "*This is similar to that reported for rainfall events in 2014 in Beijing($0.26 \times 10^6$, $0.35 \times 10^6$ and $0.14 \times 10^6$ for SNA) by Xu et al. (2017) and within the range of those estimated in the eastern United States ($0.11$-$0.38 \times 10^6$, $0.38$-$0.97 \times 10^6$ and $0.2$-$0.75 \times 10^6$ for SNA) (Hicks, 2005). Although the W ratios in this study are the same magnitude as the previous studies, some difference still exist*".

*5. It will be better to unify the wording of "washout/rainout" or "below-cloud/in-cloud scavenging" throughout manuscript.*

**[Response]:** Thank you for the suggestion. All the description of "washout/rainout" have been changed as the "below-cloud/in-cloud scavenging" throughout manuscript.

References:

Li, C., Joiner, J., Krotkov, N. A., and Bhartia, P. K.: A fast and sensitive new satellite SO2 retrieval algorithm based on principal component analysis: Application to the ozone monitoring instrument, Geophys Res Lett, 40, 6314-6318, 10.1002/2013gl058134, 2013.

Tang, W. F., Arellano, A. F., Gaubert, B., Miyazaki, K., and Worden, H. M.: Satellite data reveal a common combustion emission pathway for major cities in China, Atmos Chem Phys, 19, 4269-4288, 10.5194/acp-19-4269-2019, 2019.

Torres, O., Bhartia, P. K., Herman, J. R., Sinyuk, A., Ginoux, P., and Holben, B.: A long-term record of aerosol optical depth from TOMS observations and comparison to AERONET measurements, J Atmos Sci, 59, 398-413, 2002.

Zhang, L. S., Lee, C. S., Zhang, R. Q., and Chen, L. F.: Spatial and temporal evaluation of long term trend (2005-2014) of OMI retrieved NO2 and SO2 concentrations in Henan Province, China, Atmos Environ, 154, 151-166, 10.1016/j.atmosenv.2016.11.067, 2017.

---

## Author Comment (AC2) · 6 Mar 2021

The authors appreciate the reviewers for reviewing our manuscript and providing constructive comments. As suggested, we carefully revised the manuscript thoroughly according to the valuable advices, as well as the typographical, grammatical, and bibliographical errors. Listed below are our point-by-point responses in blue to the review's comments (in italic).

**Anonymous Referee #2**

*This study analyzed concentrations of sulfate, ammonium, nitrate and other inorganic ions in precipitation and fine PM and estimated the fraction of the wet deposition attributed to in-cloud and below-cloud scavenging processes. The sequential sampling of precipitation over the course of a precipitation event provides interesting insight into changes in the precipitation chemistry that cannot be obtained from daily or weekly precipitation monitoring that are reported in many studies. The study introduced a newer approach to estimate the in-cloud and below-cloud wet scavenging proportions. In spite of the novel aspects in this study, the scientific discussions on wet scavenging of acidifying pollutants needs to be improved.*

**[Response]:** We appreciate the valuable comments of the anonymous referee. We have prepared the point-by-point responses to address the reviewer's comments as shown below.

*The results and discussion center on the wet scavenging of inorganic ions in PM2.5. There was hardly any discussion on the wet scavenging of coarse particles and gas phase S and N compounds that can also contribute substantially to wet deposition of sulfate, nitrate and ammonium. The paper discussed relationships between precipitation and ambient air concentrations of inorganic ions; however, the latter was based on components in PM2.5. How would the relationship change if coarse PM and gas phase compounds were included? Similarly for the below-cloud scavenging results, there needs to be a more balanced discussion on particle and gas-phase scavenging.*

**[Response]:** Thanks for the comments. We are absolutely agreed with that the other sources, i.e., coarse particles and gas phase S and N compounds, can contribute to wet deposition of sulfate, nitrate and ammonium. Evenly, the model study in Japan by Kajino et al. (2015) showed the $SO_2$ and $HNO_3$ gases dominantly contributed to the washout of $SO_4^{2-}$ and $NO_3^-$, while the aerosol removal was dominated by the rainout process. However, in our measurement study, the on-line observation of coarse particles and gas phase S and N compounds were absent. We will try to address this in the revised manuscript and even in our future research.

1. The results in Kobe (high emission flux of $SO_2$ in Japan) showed the large difference between the observation and simulation, mostly due to the absent consideration of gas dissolved into raindrop in high-frequency observation measurements (Kajino et al., 2015). In their model set up, all of below-cloud gas $SO_2$ was assumed to be dissolved into raindrop and be fully oxidized to $SO_4^{2-}$. However, as suggested by Seinfeld and Pandis (2006), the aqueous equilibrium

between ambient gas and precipitation cannot be assumed due to the relatively short residence times of falling precipitation. Considering the large amounts of particles (60-90 µg/m$^3$ in mass concentration) below-cloud in Beijing, the gases compounds may be not as important as that in simulation in Japan. According to the yearly book of "*Environmental Bulletin in Beijing*" from 1994 to 2017, the yearly concentration of SO$_2$ has a dramatically decreasing from 26.5 µg/m$^3$ in 2013 to 8 µg/m$^3$ in 2017. This relatively low-level concentration of SO$_2$ at surface may not contribute a dominant role in wet deposition of SO$_4^{2-}$. Similar case in NO$_3^-$, the ratio of gas-phase HNO$_3$ and the total NO$_3^-$ in the summer in Beijing is only 0.12 according to the measurement study of Yue *et al.* (2013). We assumed the 10% ratio of gases added into the washout process, which only leads to less 5% difference of below-cloud scavenging contribution to total wet depositions. Anyway, for NH$_3$, there might be larger uncertainties, since the high concentration of NH$_3$ at below-cloud layer over NCP (Pan et al., 2017). Kasper-Giebl et al. (1999) reported that 49-79% of NH$_4^+$ in precipitation are from particulate ammonium, which indicate the large uncertainties of contribution from gases still exists in the form of NH$_4^+$ wet deposition. This also confirmed by our results: the large difference in the below-cloud contribution to NH$_4^+$ wet deposition estimated by the exponential approach and the average approach in Table 1 of the revised manuscript. Thus, the more research on the effect of NH$_3$ to NH$_4^+$ wet deposition in Beijing should be considered in the future.

2. As to particles wet scavenging, the main factors affecting the below-cloud scavenging include raindrop number size distribution, collection efficiency and raindrop terminal velocity. For collection efficiency, Brownian diffusion, directional interception, inertial impaction, thermophoresis and diffusion electrophoresis are the critical affected mechanisms. Coarse particles (aerosol particle sizes ranging from 2-20 µm) are easily scavenged by inertial impaction. Especially coarse particles (> 20 µm) are also easily scavenged through the effect of gravity. Fine particles (< 0.2 µm) can be removed by Brownian diffusion. However, accumulation mode aerosols (0.2 µm-2 µm) are neither efficiently scavenged by Brownian diffusion nor by directional interception or inertial impaction. In addition, phoretic and electric charging effects mainly affected the particle size range of 0.2- 3 µm. And electrical effect is also one of undeniably mechanisms. Since the ions collected in precipitation are both from fine and coarse particles, the effect of coarse particles in wet deposition is considered in this study. However, as to the analysis of relationship between precipitation and ambient air concentrations of inorganic ions, the effects of coarse particles are not considered. This may bring little influence to SO$_4^{2-}$ and NH$_4^+$, but large uncertainties in NO$_3^-$, due to their particle size distribution characteristics. The lower coefficient in NO$_3^-$ than SO$_4^{2-}$ and NH$_4^+$ may also be attributed to this reason.

*A more detailed explanation as to why the beginning and latter precipitation fractions represent below-cloud and in-cloud scavenging, respectively, is necessary since most of the results from this study are based on this key assumption. This data analysis*

*method seems to assume that below-cloud and in-cloud scavenging have to occur in sequence.*

**[Response]:** Thanks for the comment. The data analysis method in this study do not assume that the below-cloud and in-cloud scavenging occur in sequence. Both of the below-cloud and in-cloud scavenging contribute to the total wet deposition (Seinfeld and Pandis, 2006). The general measurement to precipitation monitoring at ground surface cannot distinguish the two processes from each other as they are already mixed well in the raindrop falling down to the ground. However, according to (Seinfeld and Pandis, 2006), species can be incorporated into cloud and raindrops inside the raining cloud and this process determine the initial concentration of raindrops before they start falling below the cloud base. In this stage, despite of the efficient process of the nucleation scavenging in cloud, the total mass of aerosol in cloud is almost stable due to the slow process of interstitial aerosol collection by cloud droplets which is the determination process to aerosol mass. That is to say, the initial concentration of raindrops in cloud is well mixed and can be considered as a stable statue during the whole rainfall event. Actually, many observations in different regions (Aikawa *et al*., 2009; 2014; Wang et al., 2009; Quyang et al., 2015; Xu et al., 2017) reported that the chemical components in a rainfall event show a decayed variation with the increase of precipitation amount and eventually tends to a stable and low concentration level. The assumption in this study as well as the previous studies "*the concentrations in later increments can be attributed to scavenging by rainout only*" is based on this fact. It does not mean the below-cloud and in-cloud scavenging occur in sequence. But, instead, the two processes have been mixed in all stage of the rainfall event with the below-cloud scavenging contributed more in beginning fraction and the in-cloud scavenging contributed more in the later fraction due to the depletion of the air pollutants below cloud by washout. The question is, how we recognized the "later fraction" in a rainfall event. The 5 mm accumulated precipitation (the concentration of chemical components at the 5[th] fraction) was used in the previous studies. However, in many cases, the concentration of chemical ions after 5 mm accumulated precipitation cannot be reached at stable level due to different precipitation intensity and the concentration level of air pollutants in each rainfall event. In this study, the asymptote value from the exponential decay fitting curve of the observed rainwater concentrations was employed as the in-cloud ion concentration. To investigate the uncertainties of this approach, the comparison between the exponential approach and the average value in fractions 6 to 8 (here after, average approach) has been carried out. The results show the exponential approach gives lower estimates of in-cloud concentrations than the average approach, with the latter being recognized as an upper limit for in-cloud concentrations. The more detailed explanation on the estimation of below cloud scavenging has been added in the revised manuscript.

*The discussion on temporal variations in precipitation concentration of inorganic ions is lacking some depth. Further analysis with annual emissions data are necessary to explain interannual changes and appreciate the impact of the air pollution control policies implemented.*

**[Response]:** Agree. Further analysis on the temporal variations in precipitation

concentration of inorganic ions with change of annual emissions have been discussed in the revised manuscript. More detailed description can be seen in the response to the comments "L185" and "section 3.1".

*The way the below-cloud percentages are reported in the paper raises some critical questions. In this version, only the median below-cloud percentage were reported. It is uncertain how representative are the median values considering that each precipitation event is unique in terms of rainfall intensity, precipitation form (rain, snow, or sleet), air concentrations of chemical components, type of weather system, etc. It is important to state or discuss under what conditions apply to those below-cloud percentages and their limitations. A range of the below-cloud percentages should also be included as well as some discussion on the variability of the below-cloud scavenging percentages.*
**[Response]:** Thanks for the comment. Indeed, each precipitation event is unique in terms of rainfall intensity, rainfall type, air concentrations of chemical components, etc. The average or median values for in-cloud concentration during 4 years cannot represent each unique precipitation event. In this study, the fitting curve is implemented in each rainfall event during 2014-2017. The median below-cloud percentage in Figure 1 is used to compare with the method that reported in previous studies (Aikawa et al., 2009;2014; Xu el al., 2017), in which the median value after 5 mm accumulated precipitation over year-round measurements were recognized as the in-cloud concentration. The comparison between the exponential approach and the average approach (average value after 5$^{th}$ fraction, i.e., 6$^{th}$-8$^{th}$) has been carried out. The results show the exponential approach gives similar ratio as the average approach for most ions, except for $NH_4^+$, $F^-$, $K^+$ and $Mg^{2+}$, where the maximum difference is less than 20% (Table 2). In general, the exponential approach gives lower estimates of in-cloud concentrations than the average approach. The latter may be recognized as an upper limit for in-cloud concentrations.

Based on this uncertainty analysis, the exponential approach to each unique event was made. Yearly ratio of below-cloud contribution to total wet deposition were calculated according to Eq (1-2), in which the $\bar{C}$ is based on the median value of in-cloud concentration in all events in each year. This is consistent with that reported in (Aikawa et al., 2009), who also used the median value as the yearly in-cloud concentration. Besides, the estimations based on the average value as well as the first quartile and the third quartile are also included in the Figure 4 of revised manuscript. It is shown that the overall trend is the same as the median value. Thus, the conclusion is more robust through comparing with the different $\bar{C}$ values.

To avoid the misunderstanding, the clarify of the method for estimating the below-cloud proportion has been added in the revised manuscript, which is as: *Noted that, the fitted exponential curve is applied to the combination of all 69 full events to estimate the yearly median concentration of chemical ions in-cloud and to compare with the results from previously reported method (i.e., median concentration after 5 mm increments). Besides, the exponential approach to each unique event was also employed.*

$$\text{Wetdep}_{below-cloud} = \sum_{i=1}^{n}(C_i - \bar{C}) \times P_i \qquad (1)$$

$$\text{Contribution}_{below-cloud} = \frac{\text{Wetdep}_{below-cloud}}{\sum_{i=1}^{n} C_i \times P_i} \qquad (2)$$

Besides, the sentence in L302 of the original manuscript has been revised as: *The ratio of below-cloud contribution to the four major components based on the yearly median value of the in-cloud concentration is also shown in Figure 4* in the revised version.

[Figure]

Figure 4 (in the revised manuscript). The annual volume weighted average below-cloud and in-cloud portion of $SO_4^{2-}$ (a), $Ca^{2+}$ (b), $NO_3^-$ (c), and $NH_4^+$ (d) during 2014-2017. The ratio of annual median below-cloud contribution for each component is represented as the black line in each panel. The mark #M and #A in the ratio of below-cloud represent the estimation based on the median value and average value of in-cloud concentration in each year, while the first quartile and the third quartiles are also included in the figure.

***Specific comments:***
*L22-24: Why would the Action Plan result in declines in Ca2+ given that it is mainly derived from crustal emissions?*

**[Response]:** The Action Plan launched in 2013 including emission reduction not only from energy consumption of industry but also the raised dust in cities. This should result the decline in $Ca^{2+}$. Unfortunately, both of the emission and the concentration data of $Ca^{2+}$ are absent in this study. Instead, the different of $PM_{10}$ and $PM_{2.5}$ ($PM_{10}$-$PM_{2.5}$) air concentration during 2013-2017 have been calculated to represent the coarse particles, which contains the $Ca^{2+}$ compound. The result shows that the concentration decreased from 31.2 µg/m$^3$ in 2013-2014 to 24.0 µg/m$^3$ over 2015-2017. This indicates the improvement of coarse particles even which is derived from crustal emissions have been made through the Action Plan launched in 2013 in China.

*L25: "An improved sequential sampling method: : :" This is vague. The improvement to the method needs to be clarified in the abstract. It is not the improvement to the*

sequential sampling method that is described in the paper, but rather the improvement to the method of estimating the below-cloud scavenging proportion.

**[Response]:** Thanks for the comment. The sentence has been revised as: *An improved method of estimating the below-cloud scavenging proportion based on sequential sampling is developed and implemented to estimate the contribution of below-cloud and in-cloud wet deposition over the four-year period.*

*L27: Suggested revision, "below-cloud scavenging accounts for one half to two thirds of wet deposition". To be more concise, only the second part of the sentence is necessary because it is very similar to the first part of the sentence.*

**[Response]:** Agree. Considering the uncertainties of below-cloud scavenging contribution to $NH_4^+$ wet deposition, the sentence has been revised as*: Overall, the below-cloud scavenging plays a dominant role to the wet deposition of four major ions at the beginning of the Action Plan.*

*L30: Was there a continuous year-to-year decline in the percentages, i.e. in 2015 and 2016? If so, this should be stated. If not, the next sentence, which suggests the result is due to the Action Plan, should not be included in the abstract because there is no clear decreasing trend.*

**[Response]:** Thanks for the comment. No, there is not a continuous year-to-year decline in the percentages. The percentages from 2015 to 2016 show a little increasing change. Nevertheless, a decreasing trend, on the whole, have been observed in S and N compound both in airborne and precipitation after the Action Plan launched in 2013. All of the ions in precipitation except $NO_3^-$ show a significant decrease (p>0.01). Thus, the expressions on the Action Plan in the next sentence are kept in the abstract.

*L36-37: ": : :clearly identifies oxidized nitrogen species as a major target for future air pollution controls." This statement seems to be based only on the lack of declining trends in airborne and precipitation nitrate. The observed concentrations of nitrate also need to be considered compared to other N or S compounds. In precipitation, it seems that the VWA of nitrate was similar to that of sulfate and ammonium was even higher than nitrate. It is not clear why the policy recommendation is to target oxidized nitrogen and not other N or S compounds as well.*

**[Response]:** Thanks for the comment. We agree with that the other N and S compounds should be paid continuous attention in the future. However, the decreasing from both VCD $NO_2$ and nitrate is absent, while declining for S is significant. Besides, the statement is not only based on the lack of declining trend in airborne and precipitation nitrate, but also the decreasing ratio of S/N in both precipitation ($SO_4^{2-}/NO_3^-$, μeq/L) and air ($SO_4^{2-}/NO_3^-$, μg/m$^3$) as well as the decreasing ratio of $NH_4^+/NO_3^-$ from 2014 to 2017. These evidences clearly confirm that nitrate should be the major target for air pollution controls in the next Action Plan.

*L44-46: Do these factors affect below-cloud rain scavenging only or both rain and snow?*

**[Response]:** Yes, these factors affect both of the rain and snow through below-cloud scavenging. However, in this study, the measurement is only focused on rain instead of snow. The sentence has been revised as: *The below-cloud scavenging process depends both on the characteristics of the rain (snow).*

*L57-58: For which chemical components do these results apply to? Can you provide details on the modeled and measured scavenging coefficients?*

**[Response]:** Yes, we can provide details on the modeled and measured scavenging coefficients. Besides, the below-cloud scavenging coefficients used in CTM is referred to $PM_{2.5}$, while in observations is referred to SNA. The expression here is revised as: *They found that below-cloud scavenging coefficients for $PM_{2.5}$ widely used in CTMs ($\sim 10^{-5}$-$10^{-6}$) were 1-2 orders of magnitude lower than estimates from observations (at the range of $10^{-4}$-$10^{-5}$ for $SO_4^{2-}$, $NO_3^-$ and $NH_4^+$, respectively).*

*L60-64: Here you argue that uncertainties in below-cloud scavenging coefficients is a potential reason for model underestimation of nitrate and sulfate wet deposition. How important are other sources of model uncertainties, such as N and S emissions, chemical transformation, changes in other ambient N and S compounds, etc.?*

**[Response]:** Thanks for the comment. The other sources of model uncertainties, such as N and S emissions, chemical transformation, changes in other ambient N and S compounds, are also important to the simulation of nitrate and sulfate wet deposition. It is revised as: *This could be one reason for the underestimation of $SO_4^{2-}$ and $NO_3^-$ wet deposition in regional models of Asia reported in phase II and III of the Model Inter-Comparison Study for Asia (MICS-Asia) (Wang et al., 2008; Itahashi et al., 2020; Ge et al., 2020) and in global model assessments by the Task Force on Hemispheric Transport of Atmospheric Pollutants (TF-HTAP) (Vet et al., 2014), in addition to the other sources of model uncertainties (Chen et al., 2019; Tan et al., 2020; Kong et al., 2020), such as emissions, chemical transformation and changes in other ambient compounds of sulfur and nitrogen.*

*L78-80: "The chemical components in later increments of rainfall are thought to be less influenced by the below-cloud scavenging process than by the in-cloud rainout process". The wording is confusing. Why does the latter precipitation increments in sequential sampling represent in-cloud scavenging, while the start of the precipitation increments represent below-cloud scavenging? At what point during a precipitation event does the dominant wet scavenging process change from below-cloud to in-cloud? Is this transition point the same for all precipitation events and locations? Each precipitation event is unique in terms rainfall intensity, droplet sizes and distribution, precipitation form (rain, snow, or sleet), air concentrations of chemical components, etc. Meteorology varies with location; some locations are frequently affected by thunderstorms or deep convective scavenging. Can you address whether these factors are taken into consideration when using the sequential precipitation sampling method to estimate incloud or below-cloud scavenging contributions?*

**[Response]:** Thanks for the comment. Similar as the response to the general comments

2 and 4. The initial concentration of raindrops in cloud is well mixed and can be considered as a stable statue during the whole rainfall event. Actually, many observations in different regions (Aikawa *et al*., 2009; 2014; Wang et al., 2009; Quyang et al., 2015; Xu et al., 2017) reported that the chemical components in a rainfall event show a decayed variation with the increase of precipitation amount and eventually tends to a stable and low concentration level. The assumption in this study as well as the previous studies "*the concentrations in later increments can be attributed to scavenging by rainout only*" is based on this fact. It does not mean the below-cloud and in-cloud scavenging occur in sequence. But, instead, the two processes have been mixed in all stage of the rainfall event with the below-cloud scavenging contributed more in beginning fraction and the in-cloud scavenging contributed more in the later fraction due to the depletion of the air pollutants below cloud by washout.

The question is, how we recognized the "later fraction" in a rainfall event. The 5 mm accumulated precipitation (the concentration of chemical components at the $5^{th}$ fraction) was used in the previous studies. However, in many cases, the concentration of chemical ions after 5 mm accumulated precipitation cannot be reached at stable level due to different precipitation intensity and the concentration level of air pollutants in each rainfall event. In this study, the asymptote value from the exponential decay fitting curve of the observed rainwater concentrations was employed as the in-cloud ion concentration. To investigate the uncertainties of this approach, the comparison between the exponential approach and the average value in fractions 6 to 8 (here after, average approach) has been carried out. The results show the exponential approach gives lower estimates of in-cloud concentrations than the average approach, with the latter being recognized as an upper limit for in-cloud concentrations.

And indeed, each precipitation event is unique in terms of rainfall intensity, rainfall type, air concentrations of chemical components, etc. The exponential approach to each unique event was made. In this sense, the unique meteorology conditions as well as air pollutions are considered in each rainfall event, since the variations of the ions concentration in each fraction of each rainfall are influenced by these synthetic effects. Besides, the estimation of below-cloud scavenging proportion based on the average value as well as the first quartile and the third quartile value are also included in the Figure 4 in addition to the median value to show a robust decreasing trend of below-cloud scavenging in wet deposition during 2014-2017. These have been revised in the new version.

*L105: How much precipitation is collected in the eighth fraction? These details should be included in the paper.*
**[Response]:** Agree. The eighth fractions in each rainfall event are different. The detail information has been added in the revised manuscript as: *During 2014-2017, a total of 104 precipitation events, which is almost 690 precipitation samples, were collected. Of the total number of precipitation events, 33 events (32%) were discarded from the sequential sampling analysis due to low rainfall amounts (<8 mm), which cannot satisfy the rules of full events. Altogether, 69 full events including 6 extended events were recorded over the 2014-2017 period in Beijing, as 15, 16, 20 and 18 events at each year,*

*respectively. The rainfall volume of the eighth fraction of these 69 full events varied from 1 mm to 55.9 mm.*

*L109: In the 69 full events, what were the total precipitation amounts collected? Are they roughly the same for each event or do they vary greatly? At what point do you have to conduct manual sampling to collect the remainder of the precipitation? These details should be included in the paper.*

**[Response]:** Agree. Similar as the response to major points 1 of the Anonymous Referee #1. During 2014-2017, a total of 104 precipitation events, which is almost 690 precipitation samples, were collected. Of the total number of precipitation events, 33 events (32%) were discarded from the sequential sampling analysis due to low rainfall amounts (<8 mm), which cannot satisfy the full events. The total precipitation amounts over 2014-2017 were 2151 mm with the data in each year listed in Table S1. The rainfall volume varied greatly for each event. Since mainly precipitation events occurred in night. The manual sampling has only been conducted in several heavy rainfall events which occurred in daytime. More details introduction on sampling have been added in the revised version as: *For example, if there is 12 mm rainfall volume in a precipitation event, 1 mm sequential rainfall is collected in each of the first 7 fractions with the rest of 5 mm in the eighth fraction. Rainfall events where eight fractions are collected and identified as full events, and those with fewer than eight fractions are characterized as incomplete events. Manual sampling methods were used to collect more than eight fractions during heavy rainfall, and these are characterized as extended events. During 2014-2017, a total of 104 precipitation events, which is almost 690 precipitation samples, were collected. Of the total number of precipitation events, 33 events (32%) were discarded from the sequential sampling analysis due to low rainfall amounts (<8 mm), which cannot satisfy the rules of full events. Altogether, 69 full events including 6 extended events were recorded over the 2014-2017 period in Beijing, as 15, 16, 20 and 18 events at each year, respectively. The rainfall volume of the eighth fraction of these 69 full events varied from 1 mm to 55.9 mm.*

*L141-147: Is there additional evidence that can be presented to justify the assumption that precipitation at the start of an event represents below-cloud scavenging while latter precipitation represents in-cloud scavenging? Some studies have measured concentrations of chemical components in cloud water as an indication of in-cloud scavenging; however, the method in this study are based solely on the precipitation at the surface. One initial criticism I have on this assumption is why below-cloud and in-cloud processes have to occur one after another instead of simultaneously. A full justification is necessary given that the major results of this study depend on this analysis and interpretation.*

**[Response]:** Thanks for the comment. According to the reported concentrations of chemical components in cloud water (Seinfeld and Pandis, 2006), which is 4-5 times higher than that observed at the surface. However, this can not be an indication of in-cloud scavenging, since the concentration might be diluted as the rainfall occurred. The direct observation just under the base of the cloud should be the concentration in-cloud.

Unfortunately, this measurement was absent in this study.

Similar as the response to the general comments 2. The assumption in this study does not mean the below-cloud and in-cloud scavenging occur in sequence. But, instead, the two processes have been mixed in all stage of the rainfall event with the below-cloud scavenging contributed more in beginning fraction and the in-cloud scavenging contributed more in the later fraction due to the depletion of the air pollutants below cloud by washout. The more detailed explanation on the estimation of below cloud scavenging including the full justification has been added in the revised manuscript.

*L157: In previous studies, it seems that the 5 mm accumulated precipitation cut-off was based on the point where there was a lack of change in the precipitation concentrations. This is not entirely subjective.*

**[Response]:** Agree. The expression has been revised as: *Previous studies have estimated the concentration of chemical ions scavenged in-cloud based on the judgment that 5 mm of accumulated precipitation is sufficient to identify the contribution of the rainout process (Wang et al., 2009;Aikawa and Hiraki, 2009;Xu et al., 2017).*

*L164-174: Clarity on the methodology is needed. You mentioned that each precipitation event is unique in terms of the decreasing rate in the precipitation concentrations. What is the reason for combining all the precipitation events prior to fitting a regression curve? Why not fit a curve for each event? What is the purpose of fitting the data through the first and third quartiles – can this serve as an estimate of the uncertainties? "In theory, the concentration of chemical ions stabilize at higher rainfall increments and this represents the concentration in cloud." A brief explanation of the theory should be provided in the paper.*

**[Response]:** Agree. In this study, the fitting curve is implemented both in each rainfall event and in all combining events over 2014-2017. The latter is used to compare with the yearly scale in-cloud concentration which is reported in previous study from the median data. The first and third quartiles served as an estimate of the range of the uncertainties. The comparison between the exponential approach and the average approach (average value after $5^{th}$ fraction, i.e., $6^{th}$-$8^{th}$) has been carried out. The results show the exponential approach gives lower estimates of in-cloud concentrations than the average approach, with the latter being recognized as an upper limit for in-cloud concentrations. Based on this uncertainty analysis, the exponential approach to each unique event was made. This methodology has been revised to more clarity to the readers. It was now revised as: *In this study, a new method based on fitting a curve to the chemical ion concentrations with successive rainfall increments has been developed to estimate the contribution of the rainout process. As shown in Figure 1, an exponential curve is fitted to the median, $25^{th}$ and $75^{th}$ percentiles of the chemical ion concentrations in each fraction through the rainfall increments. Noted that, the fitted exponential curve is applied to the combination of all 69 full events to estimate the yearly median concentration of chemical ions in-cloud and to compare with the results from previously reported method (i.e., median concentration after 5 mm increments). Besides, the exponential approach to each unique event was also employed. Ideally, the*

*concentration of chemical ions stabilize at higher rainfall increments and this represents the concentration in cloud. However, the decrease during each rainfall event is distinctly different, and this regression method is not fully applicable to all rainfall events in practice. Therefore, the exponential regression method is used to estimate the in-cloud concentration under most circumstances, but where the decreasing trend with the increment of rainfall is not significant, the average value of rainfall increments 6-8 of the event is used.*

*Equations 1,2: What does n represent? n in eq. 1 is different from the n in eq. 2.*
**[Response]:** n represent the total fractions in a rainfall event. This has been added in the revised manuscript.

*L185: Some background information on the Action Plan is needed in the introduction to understand what effects these policies may have on air quality. Which pollutants does the Action Plan target and what was implemented?*
**[Response]:** The Action Plan is launched in 2013 called "Ten rules" to improve the air quality in China. It includes comprehensive control of industrial emission, non-point emission, fugitive dust, vehicles, etc. It is also aimed to adjust and optimize the industrial structure and promote economic transformation and upgrading, such as increase the supply of clean energy. These actions are ensured to work by both of legislation and market mechanism. According to the *Beijing Environmental Statement* published by the Beijing Municipal Environmental Protection Bureau from 2013 to 2017, many measures have been implemented to meet the Action Plan, including replacement residential coal with electricity and natural gas, upgrading the emission standards of gasoline, diesel vehicles and power plants, closing the high-emission enterprises. This description has been added in section 3.1 of the revised manuscript.

*Section 3.1: Why are the results of other inorganic ions not discussed? The annual changes in the VWM concentrations should be discussed with changes in emissions of the precursor pollutants, such as SOx and NOx emissions, and policies that were introduced to manage those emissions. There should also be more discussion on what was causing the significant declines in sulfate and Ca2+ in the earlier period. Air pollution control measures typically target major anthropogenic contaminants, but here you have shown significant decreases in Ca2+ which is largely from soil emissions. It is not clear how the Action Plan played a role in the reduction of VWM Ca2+. For a better understanding of the impacts of acidification on ecosystems, wet deposition fluxes should be presented as well. While there may be significant decreases in VWM concentrations, interannual variability in precipitation amounts may result in less significant decreases in wet deposition fluxes.*
**[Response]:** In this study, the major ions in precipitation are considered. According to the measurement, the two anions ($SO_4^{2-}$ and $NO_3^-$) and two cations ($NH_4^+$ and $Ca^{2+}$) are discussed in the main text with the other ions listed in Table 2 and plotted in Figure 7 in the revised manuscript.

The annual changes in VWM concentrations with changes in emissions and the

annual concentrations of $SO_2$ and $NO_x$ have been added in section 3.1 of the revised manuscript. The emission and the concentration data of $SO_2$ and $NO_x$ are collected from the yearly book of "*Environmental Bulletin in Beijing*" from 1994 to 2017. It is clearly shown the concentration of $SO_2$ experienced a sustainably decreasing trend due to significant reduction of its emission from 1996 to 2017, with the decreases rate is 4.5% $yr^{-1}$ and 13.9% $yr^{-1}$ in emission and 2.8% $yr^{-1}$ and 14.0% $yr^{-1}$ in concentration during 1995-2010 and 2013-2017 (the Action Plan period), respectively. This is consistent with the annual changes of VWM concentrations of $SO_4^{2-}$. As to NOx emission, the data in recent years have been collected. Although the clearly reduction is found in the annual changes of emission from 2010, the ambient concentration of $NO_2$ do not show a significant decreasing trend (~3.6% $yr^{-1}$) compared with $SO_2$ (14% $yr^{-1}$). However, before the Action Plan, the decreasing ratio in concentration is only 1.8% $yr^{-1}$, which is slower than the Action Plan period. Different from its variation in the ambient concentration, the increase of VWM concentration of $NO_3^-$ (+4% $yr^{-1}$) in precipitation is found during 1995-2010 and then decrease (-3% $yr^{-1}$) in 2013-2017. Both of the annual changes in $SO_2$ and $NO_2$ implies the dominant contribution of the Action Plan to the air quality.

The significant declines in VWM concentration of $Ca^{2+}$ is found in precipitation with the decreases rate as 36.1% $yr^{-1}$ in 1995-2010 and 8.8% $yr^{-1}$ in 2014-2017. The emission and the concentration data of $Ca^{2+}$ are absent in this study. Instead, the different of $PM_{10}$ and $PM_{2.5}$ ($PM_{10}$-$PM_{2.5}$) air concentration during 2013-2017 have been calculated to represent the coarse particles, which contains the $Ca^{2+}$ and $Na^+$ as well. The results show that the concentration decreased from 31.2 µg/m$^3$ in 2013-2014 to 24.0 µg/m$^3$ over 2015-2017. This indicates the improvement of coarse particles even which is derived from crustal emissions have been made through the Action Plan. Similar as the response to "specific comments L185", the Action Plan launched in 2013 including emission reduction not only from energy consumption of industry but also the fugitive dust in cities, which should result the decline in $Ca^{2+}$.

Observations on S and N wet deposition (Pan et al., 2012; 2013) during 2007-2010 show the value of 21.5 kg S ha$^{-1}$ yr$^{-1}$ and 27.9 kg N ha$^{-1}$ yr$^{-1}$ (19.7 and 8.2 kg N ha$^{-1}$ yr$^{-1}$ through $NO_3^-$ and $NH_4^+$) in Beijing, respectively. Compared with those results, significant decreases (11.4 kg S ha$^{-1}$ yr$^{-1}$ and 23.6 kg N ha$^{-1}$ yr$^{-1}$) were observed in the four-years measurements during 2014-2017 in this study.

Further analysis on annual changes listed above have been added in section 3.1 of the revised manuscript.

[Figure]

Figure 3 (in the revised manuscript). Annual changes in emission and concentration of SO$_2$ and NO$_x$ in Beijing (collected from the yearly book of "*Environmental Bulletin in Beijing*" from 1994 to 2017).

*L219: H usually denotes a Henry's Law constant. It is more suitable to use W or SR to denote scavenging ratios as this is consistent with literature.*

**[Response]:** Agree. The H has been replaced by $W$ in the revised manuscript.

*L234: "while the VWA represents a greater contribution from in-cloud removal: : :" It was stated earlier that latter fractions of the accumulated precipitation represents incloud removal. Why was the VWA concentrations (which may include below-cloud and in-cloud contributions) used instead of the concentrations in the latter precipitation fractions?*

**[Response]:** It is the VWA concentrations may include below-cloud and in-cloud contributions. The relations between the concentration of ions in VWM and the ambient air shows the synthetic correlations. Based on the comparison of the relations between air and precipitation from the first fraction or from VWM, we can get not only the influences from the air pollutants below-cloud but also the uncertainties due to coarse particles and gases below-cloud (especially to HNO$_3$ and NH$_3$). The larger difference of the R coefficients for NO$_3^-$ and NH$_4^+$ than SO$_4^{2-}$ in Figure 4 (original Figure 3) may indicate the uncertainties. The former two ions are related to more complicate sources from the ambient precursors. For example, the NO$_3^-$ in precipitation is both from the fine and coarse particles (i.e., particulate NO$_3^-$) as well as the gaseous HNO$_3$, while the NH$_4^+$ in precipitation is mainly from the fine particles in addition to NH$_3$.

*L237-238: "This indicates that the concentration of chemical ions in precipitation at the start of rainfall is more greatly influenced by aerosols below the cloud." This conclusion is not correct because the scavenging of acidic gases like HNO3 and SO2 can also contribute to nitrate and sulfate in precipitation. Another issue with the correlation analysis between precipitation and air concentrations (in Fig. 3) is that only*

*fine aerosols were considered. Would there be any changes to the correlation results if acidic gases and coarse aerosols were included?*

**[Response]:** Similar response to general comment 1. To avoid misunderstanding, this sentence has been revised as: *This indicates that the concentration of chemical ions in precipitation at the start of rainfall is more greatly influenced by the air pollutants below the cloud.*

*L245: The scavenging ratios of SNA do not include coarse particles since only PM2.5 chemical composition were measured. It also does not include the wet scavenging of acidic gases. This needs to be mentioned in the paper because typically scavenging ratios take into account all the chemical species in air that can undergo wet deposition. It seems that the scavenging ratios of SNA calculated in this study is not an accurate reflection of the wet scavenging efficiency of SNA.*

**[Response]:** Agree. The scavenging ratios of SNA in this study is not an accurate reflection of the wet scavenging efficiency of SNA due to lacking of consideration of coarse particles and acidic gases. Similar as the response to general comment 1, the absent measurement to coarse particles and acidic gases may lead to little influence to $SO_4^{2-}$, but large uncertainties in $NO_3^-$ and $NH_4^+$. To avoid misunderstanding, this has been clarified as: *It should be noted that the W calculated in this study is based on the fine particles in air, which may not represent the accurate reflection of the wet scavenging efficiency of SNA.*

*L264-266: The percentages should be converted to annualized percentages. Some of the time periods are much longer than others, which makes it appear that the rate of decrease is larger.*

**[Response]:** Agree. The percentages here have been revised to the annualized percentages in the revised manuscript.

*L269: "The chemical composition of precipitation is directly related to the amount of precipitation, and as a consequence it is difficult to identify inter-annual variations in chemical concentrations." This statement is incorrect. There are other factors affecting the precipitation chemistry besides the precipitation amount. Section 3.3: An exponential curve was used to fit the data points and then the horizontal asymptote was used to determine the precipitation concentration representative of rainout. One issue that I have with this approach is that there were only 11 data points or precipitation fractions. Previous studies (e.g. Aikawa and Hiraki, 2009) that conducted sequential sampling of precipitation found that the precipitation concentrations could increase in the much latter fractions of precipitation. It seems that more precipitation fractions need to be collected in order to observe what happens to the precipitation concentrations.*

**[Response]:** Agree. There are many other factors that affecting the precipitation chemistry. In this study, we also observed the increased precipitation concentrations in the latter fractions of precipitation. This may due to the meteorological conditions, i.e., rainfall type, intensity, cloud and the air pollution movement. Previous studies (e.g.

Aikawa and Hiraki, 2009) that conducted sequential sampling of precipitation were based on 0.5 mm increment and found the low level concentration remained (in-cloud concentration) at after 10 fractions, which is 5 mm accumulated rainfall. In this study, the sequential sampling of 1 mm increment with 8 fractions at least is employed, which covering higher than 5 mm accumulated rainfall. However, despite the longer precipitation fractions in this study were collected, the longer fraction measurement and more detailed analysis on the uncertainties are needed in the future. In this study, we agreed with the reviewer's comment and deleted this sentence in the revised manuscript.

*L303-308: "Benefiting from the Action Plan, the below-cloud contributions: : :" This part may not be necessary because it was discussed earlier that the policies in the Action Plan help reduced the VWM of inorganic ions. The Action Plan did not directly affect the below-cloud scavenging process. Rather, it improved the air quality which in turn decreased the VWM concentrations.*
**[Response]:** Thanks for the comment. We still insist to keep this part in the revised manuscript based on the following reasons:

1) Yes. It was discussed earlier that the policies in the Action Plan help reduced the VWM of inorganic ions. However, the VWM of inorganic ions in precipitation has two part (i.e., in-cloud and below-cloud contribution) in general. This part gives more detailed information that the decreases of VWM may largely due to the decreases of below-cloud part.

2) We agree with that the Action Plan did not directly affect the below-cloud scavenging process. However, the air concentration at the surface layer have been significant reduced under the implementation of the Action Plan. This improved the air quality at the surface layer and in turn decreased the below-cloud scavenging in the total wet deposition.

To avoid misunderstanding, the expression here has been revised as: *Benefiting from the Action Plan, the air quality at the surface layer have been significantly improved (Zhang et al., 2019), which in turn leading to the decreases of the below-cloud scavenging. In this study, it also shows the below-cloud contributions of $SO_4^{2-}$, $NO_3^-$, $NH_4^+$ and $Ca^{2+}$ decreases from >50% in 2014 to ~40% in 2017.*

*Section 3.4: I think this section is not necessary because the findings are not substantially new from what has already been discussed in the paper. Here you are showing the effects of rainfall on ambient air sulfate. The washout effects were already discussed in Figure 1, which shows the decrease in VWM concentrations of the inorganic ions with increasing rainfall.*
**[Response]:** Agree. The section 3.4 has been removed from the revised manuscript.

*L392: "highest contribution for $NH_4^+$ at 65% and lowest for $SO_4^{2-}$ at 50%". This was stated in the conclusion and abstract, but was not elaborated in the results. I expected the highest contribution to below-cloud wet scavenging would have been from $Ca^{2+}$ or $Na^+$ given that they are predominantly associated with coarse particles and locally emitted.*

**[Response]:** Agree. Considering the uncertainties of below-cloud scavenging contribution to $NH_4^+$ wet deposition, the expression has been deleted both in abstract and in conclusion. Besides, it is interesting to note that the yearly below-cloud contribution to total wet deposition of $Ca^+$ decreased from >60% to less 40% during 2014-2017. This may due to the clean air Action Plan, which sustainably decreased the coarse particles with the different of $PM_{10}$ and $PM_{2.5}$ ($PM_{10}$-$PM_{2.5}$) decreased from 31.2 $\mu g/m^3$ in 2013-2014 to 24.0 $\mu g/m^3$ over 2015-2017. Similar as the response to "specific comments L185 and section 3.1", the Action Plan launched in 2013 including emission reduction not only from energy consumption of industry but also the fugitive dust in cities, which should result the decline in $Ca^{2+}$.

---

## Author Response (AR2)

The authors provided a thoughtful response to the major comments. However, some of the information was not incorporated into the paper. To achieve a more balanced discussion on wet deposition of S and N, it is important to discuss in the paper whether wet scavenging of coarse particles and gas-phase S and N compounds are important or not and provide justification. Also, the theoretical explanation on below-cloud and in-cloud scavenging should be included in the paper to support the method used to estimate below-cloud and in-cloud scavenging contributions. Below are some of the specific comments that were partly addressed in the responses but not in the revised paper. Providing additional explanations in the paper would help improve its scientific quality.

**[Response]:** We appreciate the valuable comments of the anonymous referee. We have prepared the point-by-point responses to address the reviewer's comments as shown below. The additional explanations were added in the revised manuscript.

Response to "*To achieve a more balanced discussion on wet deposition of S and N, it is important to discuss in the paper whether wet scavenging of coarse particles and gas-phase S and N compounds are important or not and provide justification.*"

First, since the ions collected in precipitation are both from fine and coarse particles as well as the gas phase S and N compounds, the effect of coarse particles and gas phase S and N compounds have been considered in calculating the proportions of in-cloud and below-cloud scavenging to total wet deposition. However, the proportions estimated based on measurements cannot be distinguished from the effects of particles or gaseous compounds. The model study in Japan showed consistent fractions of in-cloud and below-cloud scavenging to total wet deposition between simulated and observed values, except one site, where is the region of high emission flux of $SO_2$. In this region, the simulated below-cloud scavenging contribution was apparently greater than the observed results. Specifically, the model shows the $SO_2$ and $HNO_3$ gases dominantly contributed to the below-cloud scavenging of $SO_4^{2-}$ and $NO_3^-$ in the regions of high emission flux of $SO_2$, in while the aerosol removal was dominated by the in-cloud scavenging process. In their model set up, all of below-cloud gas $SO_2$ was assumed to be dissolved into raindrop and be fully oxidized to $SO_4^{2-}$. However, as suggested by Seinfeld and Pandis (2006), the aqueous equilibrium between ambient gas and precipitation cannot be assumed due to the relatively short residence times of falling precipitation. Thus, the assumptions used in Kajino et al. (2015) might overestimate the contribution of gas $SO_2$ to below-cloud scavenging. Besides, considering the large amounts of particles (60-90 µg/m$^3$ in mass concentration) below-cloud in Beijing, the gases compounds may be not as important as that in simulation in Japan. According to the yearly book of "*Environmental Bulletin in Beijing*" from 1994 to 2017, the yearly concentration of $SO_2$ has a dramatically decreasing from 26.5 µg/m$^3$ in 2013 to 8 µg/m$^3$ in 2017. This relatively low-level concentration of $SO_2$ at surface may not contribute a dominant role in wet deposition of $SO_4^{2-}$. Similar case in $NO_3^-$, the ratio of gas-phase $HNO_3$ and the total $NO_3^-$ in the summer in Beijing is only 0.12 according to the measurement study of Yue *et al*. (2013). The fraction of total inorganic nitrate as particulate nitrate instead of gaseous nitric acid over the NCP increased from 90% in 2013 to 98% in 2017 (Zhai et al., 2021), which means the gaseous nitric acid has been consumed by high level of ammonia concentrations. We assumed the 10% ratio of gases

added into the washout process, which only leads to less 5% difference of below-cloud scavenging contribution to total wet depositions. Anyway, for $NH_3$, there might be larger uncertainties, since the high concentration of $NH_3$ at ground surface over NCP (Pan et al., 2018). Kasper-Giebl et al. (1999) reported that 49-79% of $NH_4^+$ in precipitation are from particulate ammonium, which indicate the large uncertainties of contribution from gases still exists in the form of $NH_4^+$ wet deposition. The uncertainties are mainly from the indistinct window for the in-cloud scavenging judgement due to high concentration of gas $NH_3$ at ground surface which is not easy to be scavenged completely during the short time fraction measurements. This is also confirmed by the larger difference in below-cloud contribution to $NH_4^+$ wet deposition than other ions estimated by the exponential approach and the average approach in Table 2.

Second, in discussion the relationships between precipitation and ambient air concentrations of inorganic ions, only components in $PM_{2.5}$ are included. Due to the absent of the on-line observed coarse particles and gas phase N compounds, the relationships as well as the scavenging ratio $W$ may exist certain uncertainties. These uncertainties have been evaluated in the revised manuscript. For S, we added gas $SO_2$ to testified its role to the relationships. Figure 1 shows the Relationships between the concentration of $SO_4^{2-}$ in precipitation and in air ($SO_4^{2-}$ in precipitation vs $SO_4^{2-}$, and $SO_4^{2-}$ in precipitation vs $SO_2+SO_4^{2-}$). The correlation coefficients R increased if the role of gas $SO_2$ was considered (R of $SO_4^{2-}$ in precipitation vs $SO_4^{2-}$ is 0.7, and R of $SO_4^{2-}$ in precipitation vs $SO_2+SO_4^{2-}$ is 0.75). However, the scavenging ratio $W$ was not changed, with the difference lower that 1%. For N, the contribution of gaseous $HNO_3$ to total inorganic nitrate is less than 2% in NCP according to Zhai et al. (2021), which can be ignored in this study. According to more than one-year measurements in Beijing (Tian et al., 2016), $SO_4^{2-}$, $NO_3^-$ and $NH_4^+$ in coarse particles account for 18%, 27% and 10%, respectively. The lower coefficient R in $NO_3^-$ than $SO_4^{2-}$ and $NH_4^+$ in Figure 4 is attributed to the absent of considering $NO_3^-$ in coarse particles. Besides, due to high concentration of $NH_3$ at ground surface over NCP (Pan et al., 2018), the $NH_4^+$ in precipitation from gaseous $NH_3$ cannot be ignored (Kasper-Giebl et al., 1999). The ratio of $NH_4^+/(2SO_4^{2-}+NO_3^-)$ in precipitation and in $PM_{2.5}$ was calculated. The lower ratio in precipitation than that in $PM_{2.5}$ was found, with 0.95-1.01 in precipitation and 1.35 in air. This implicated the impacts of rich gas $NH_3$ at ground surface going into the precipitation by reacting with gaseous $HNO_3$ and forming as $NH_4NO_3$ after $(NH_4)_2SO_4$. Thus, the contribution of coarse particles and gases to the relationships of S and N compounds in precipitation and the atmosphere is not as important as the fine particles, except $NO_3^-$ in coarse particles and the gaseous $NH_3$, which should be considered in the future.

[Figure]

Figure 1. Relationships between the concentration of $SO_4^{2-}$ in precipitation and in air in the 6 h before each precipitation event. The solid (hollow) red square and blue triangle represented the relationships between the $SO_4^{2-}$ concentration ($SO_4^{2-}+SO_2$) in air with that in F1# and in VWA, respectively. The solid and hollow lines represented linear regression line of $SO_4^{2-}$ in precipitation and in air as well as that of $SO_4^{2-}$ in precipitation and $SO_4^{2-}+SO_2$ in air.

Response to "*Also, the theoretical explanation on below-cloud and in-cloud scavenging should be included in the paper to support the method used to estimate below-cloud and in-cloud scavenging contributions.*" The theoretical explanation on below-cloud and in-cloud scavenging have been included in the revised manuscript. For more details, please see the response to specific comments 4.

L22-24: Why would the Action Plan result in declines in $Ca^{2+}$ given that it is mainly derived from crustal emissions? - This needs to be explained in the paper.
**[Response]:** It has been explained in section 3.1 of the revised manuscript as: *The significant declines in VWM concentration of $Ca^{2+}$ is found in precipitation with the decreases rate as 36.1% yr-1 in 1995-2010 and 8.8% yr-1 in 2014-2017. The emission and the concentration data of $Ca^{2+}$ are absent in this study. Instead, the different of $PM_{10}$ and $PM_{2.5}$ ($PM_{10}-PM_{2.5}$) concentration during 2013-2017 have been calculated to represent the coarse particles, in which the $Ca^{2+}$ compound is mainly loaded. The results show that the concentration decreased from 31.2 $\mu g/m^3$ in 2013-2014 to 24.0 $\mu g/m^3$ over 2015-2017. This indicates the improvement of coarse particles even which is derived from crustal emissions have been made through the Action Plan. As that is mentioned above, the Action Plan including emission reduction not only from energy consumption of industry but also the fugitive dust in cities, which should result the decline in $Ca^{2+}$.*

L30: Was there a continuous year-to-year decline in the percentages, i.e. in 2015 and 2016? If so, this should be stated. If not, the next sentence, which suggests the result is

due to the Action Plan, should not be included in the abstract because there is no clear decreasing trend. - In the response, the authors stated there was an overall decline but not a year-to-year decline in the percentages. If there is no year-to-year decline, the percentage decreases from 2013 to 2017 are likely not statistically significant.

**[Response]:** Thanks for the comment. Although there is not a continuous year-to-year decline in the percentages, the statistically significant (p>0.01) decreasing trend have been observed in $PM_{2.5}$ concentration and the major ions in precipitation except $NO_3^-$ after the Action Plan launched in 2013. Thus, the expressions on the Action Plan in the next sentence are kept in the abstract.

L78-80: Each precipitation event is unique in terms rainfall intensity, droplet sizes and distribution, precipitation form (rain, snow, or sleet), air concentrations of chemical components, etc. Meteorology varies with location; some locations are frequently affected by thunderstorms or deep convective scavenging. Can you address whether these factors are taken into consideration when using the sequential precipitation sampling method to estimate in-cloud or below-cloud scavenging contributions? - These uncertainties should be discussed in the paper because it is not clear how these factors affect wet scavenging and the proportions from in-cloud and below-cloud processes.

**[Response]:** Thanks for the comment. The unique characterization of each precipitation event was considered in calculation of the proportions from in-cloud and below-cloud processes, as the exponential approach to each unique event was made. The variations of the ions concentration in each fraction of each rainfall are influenced by the synthetic effects of meteorology conditions. After the estimation of the proportions of below-cloud scavenging in each precipitation event, the uncertainties from the meteorology conditions have been discussed in section 4 "*Factors influencing below-cloud scavenging*". In the section, the influence of these factors affecting wet scavenging were investigated through the correlation analysis between below-cloud proportions with the rainfall type as well as the rainfall intensity. The below-cloud proportions varied from 20% to 80% among the 69 rainfall events. Among these rainfall events, three types of precipitation such as cold vortex, upper-level troughs and others have been classified, based on the records of synoptic system from the Beijing Meteorological Service. A high contribution from below-cloud scavenging is found for rainfall events associated with an upper-level trough, while a lower contribution during rainfall events under cold vortex conditions. In addition, the negative correlations in below cloud fraction are found for both the rainfall volume and precipitation intensity. The heavy rainfall is corresponding to the decreasing of below-cloud proportion. The more detailed explanation was made in section 4.1 and 4.2 of the paper. To make this discussion more clearly, a description was added in the beginning of section 4 in the revised manuscript. The description is as: *Each precipitation event is unique in terms rainfall intensity, droplet sizes and distribution, rainfall type (thunderstorms or deep convective scavenging), air concentrations of chemical components, etc. The unique characterization of each precipitation event was considered in calculation of the proportions from in-cloud and below-cloud processes, as the exponential approach to*

*each unique event was made. The below-cloud proportions varied from 20% to 80%*
*among the 69 rainfall events. The influence of these factors affecting wet scavenging*
*were investigated through the correlation analysis between below-cloud proportions*
*with the rainfall type as well as the rainfall intensity.*

L141-147: One initial criticism I have on this assumption is why below-cloud and in-cloud processes have to occur one after another instead of simultaneously. A full justification is necessary given that the major results of this study depend on this analysis and interpretation. – The theoretical explanation that was provided in the response should be addressed in the paper as well.

**[Response]:** Thanks for the comment. The theoretical explanation was added in the section 2.3 of the revised manuscript. The text is as: *According to (Seinfeld and Pandis, 2006), species can be incorporated into cloud and raindrops inside the raining cloud and this process determine the initial concentration of raindrops before they start falling below the cloud base. In this stage, despite of the efficient process of the nucleation scavenging in cloud, the total mass of aerosol in cloud is almost stable due to the slow process of interstitial aerosol collection by cloud droplets which is the determination process to aerosol mass. That is to say, the initial concentration of raindrops in cloud is well mixed and can be considered as a stable statue during the whole rainfall event. That is why many observations in different regions (Aikawa et al., 2009; 2014; Wang et al., 2009; Quyang et al., 2015; Xu et al., 2017) reported that the chemical components in a rainfall event show a decayed variation with the increase of precipitation amount and eventually tends to a stable and low concentration level. The assumption in this study as well as the previous studies is based on this fact. It does not mean the below-cloud and in-cloud scavenging occur in sequence. But, instead, the two processes have been mixed in all stage of the rainfall event with the below-cloud scavenging contributed more in beginning fraction and the in-cloud scavenging contributed more in the later fraction due to the depletion of the air pollutants below cloud by washout.*

Section 3.1: Why are the results of other inorganic ions not discussed? For a better understanding of the impacts of acidification on ecosystems, wet deposition fluxes should be presented as well. While there may be significant decreases in VWM concentrations, interannual variability in precipitation amounts may result in less significant decreases in wet deposition fluxes. - The paper did not discuss the results for Na+, K+, Mg2+, Cl- and F-, which were measured in the study as stated in the methodology. There was also no discussion on the wet deposition fluxes and how changes in the precipitation amounts can affect trends in the wet deposition fluxes.

**[Response]:** Thanks for the comment. In this study, only the major ions in precipitation are considered. According to the measurement, the two anions ($SO_4^{2-}$ and $NO_3^-$) and two cations ($NH_4^+$ and $Ca^{2+}$) are discussed in the main text with the other ions listed in Table 2 and plotted in Figure 7 in the revised manuscript. As for the wet deposition, the fluxes have been added in Figure 2. The corresponding description in the paper Section 3.1 is as: *For a better understanding of the impacts of acidification on ecosystems, wet*

*deposition fluxes of the four major ions in precipitation are also plotted in Figure 2. Similar variations are found as that presented in VWA of the four major ions. Observations on S and N wet deposition (Pan et al., 2012; 2013) during 2007-2010 show the value of 21.5 kg S ha$^{-1}$ yr$^{-1}$ and 27.9 kg N ha$^{-1}$ yr$^{-1}$ (19.7 and 8.2 kg N ha$^{-1}$ yr$^{-1}$ through NO$_3^-$ and NH$_4^+$) in Beijing, respectively. Compared with those results, significant decreases (11.4 kg S ha$^{-1}$ yr$^{-1}$ and 23.6 kg N ha$^{-1}$ yr$^{-1}$) were observed in the four-years measurements during 2014-2017 in this study.*

Section 3.3: An exponential curve was used to fit the data points and then the horizontal asymptote was used to determine the precipitation concentration representative of rainout. One issue that I have with this approach is that there were only 11 data points or precipitation fractions. Previous studies (e.g. Aikawa and Hiraki, 2009) that conducted sequential sampling of precipitation found that the precipitation concentrations could increase in the much latter fractions of precipitation. It seems that more precipitation fractions need to be collected in order to observe what happens to the precipitation concentrations. – Based on the responses, the authors agree that longer fraction measurements and more detailed analysis on the uncertainties are needed in the future. This should be mentioned in the paper.

**[Response]:** Agree. It was mentioned in first paragraph of section 3.3 of the revised manuscript as: *It is also important to note that the increased concentrations of ions in the latter fractions were observed in few events in this study. This may due to the unique meteorological conditions (i.e., rainfall type, rainfall intensity) and air pollutants movement during each precipitation. Thus, despite the longer precipitation fractions in this study were collected, more longer fraction measurements and more detailed analysis on the uncertainties are needed in the future. The influences of meteorological conditions (i.e., rainfall type, rainfall intensity) are discussed in section 4.* It was also mentioned in last sentence of the second paragraph of section 3.3 in the revised manuscript as: *As it mentioned above, more longer fraction measurements as well as the influence of NH$_3$ to NH$_4^+$ wet deposition are needed in the future.*

References:

Kajino, M., and Aikawa, M.: A model validation study of the washout/rainout contribution of sulfate and nitrate in wet deposition compared with precipitation chemistry data in Japan, Atmos Environ, 117, 124-134, 10.1016/j.atmosenv.2015.06.042, 2015.

Kasper-Giebl, A., Kalina, M. F., and Puxbaum, H.: Scavenging ratios for sulfate, ammonium and nitrate determined at Mt. Sonnblick (3106 m asl), Atmos Environ, 33, 895-906, 1999.

Pan, Y. P., Tian, S. L., Zhao, Y. H., Zhang, L., Zhu, X. Y., Gao, J., Huang, W., Zhou, Y. B., Song, Y., Zhang, Q., and Wang, Y. S.: Identifying Ammonia Hotspots in China Using a National Observation Network, Environ Sci Technol, 52, 3926-3934, 10.1021/acs.est.7b05235, 2018.

Seinfeld, J. H., and Pandis, S. N.: Atmospheric chemistry and physics: from air pollution to climate change, Wiley, New York, 2006.

Tian, S. L., Pan, Y. P., and Wang, Y. S.: Size-resolved source apportionment of particulate matter in urban Beijing during haze and non-haze episodes, Atmos Chem Phys, 16, 1-19, 10.5194/acp-16-1-2016, 2016.

Yue, D., Hu, M., and Wu, Z.: Variation and interaction of major azotic inorganic compounds in the summer in beijing (in Chinese), Environ. Monit. China, 29, 9-14, 2013.

Zhai S., D. J. Jacob, X. Wang, Z. Liu, T. Wen, V. Shah, K. Li, J. Moch, K. H. Bates, S. Song, L. Shen, Y. Zhang, G. Luo, F. Yu, Y. Sun, L. Wang, M. Qi, J. Tao, K. Gui, H. Xu, Q. Zhang, T. Zhao, H. C. Lee, H. Choi, H. Liao, Control of particulate nitrate air pollution in China, Nature Geoscience, in press, 2021.